# T-Stitch: Accelerating Sampling in Pre-trained Diffusion Models with Trajectory Stitching

## Abstract

Diffusion probabilistic models (DPMs) achieve great success in generating high-quality data such as images and videos. However, sampling from DPMs at inference time is often expensive for high-quality generation and typically requires hundreds of steps with a large network model. In this paper, we introduce sampling Trajectory Stitching (**T-Stitch**), a simple yet efficient technique to improve the generation efficiency with little or no loss in the generation quality. Instead of solely using a large DPM for the entire sampling trajectory, T-Stitch first leverages a smaller DPM in the initial steps as a cheap drop-in replacement of the larger DPM and switches to the larger DPM at a later stage. The key reason why T-Stitch works is that different diffusion models learn similar encodings under the same training data distribution. While smaller models are not as effective in refining high-frequency details in later denoising steps, they are still capable of generating good global structures in the early steps. Thus, smaller models can be used in early steps to reduce the computational cost. Notably, T-Stitch does not need any further training and uses only pretrained models. Thus, it can be easily combined with other fast sampling techniques to obtain further efficiency gains across different architectures and samplers. On DiT-XL, for example, 40% of the early timesteps can be safely replaced with a 10x faster DiT-S without performance drop on class-conditional ImageNet generation. By allocating different fractions of small and large DPMs along the sampling trajectory, we can achieve flexible speed and quality trade-offs. We further show that our method can also be used as a drop-in technique to not only accelerate the popular pretrained stable diffusion (SD) models but also improve the prompt alignment of stylized SD models from the public model zoo.

## 1 Introduction

Diffusion probabilistic models (DPMs) (Ho et al., 2020) have demonstrated remarkable success in generating high-quality data among various real-world applications, such as text-to-image generation (Rombach et al., 2022), audio synthesis (Kong et al., 2021) and 3D generation (Poole et al., 2023), etc. Achieving high generation quality, however, is expensive due to the need to sample from a large DPM, typically involving hundreds of denoising steps, each of which requires a high computational cost. For example, even with a high-performance RTX 3090, generating 8 images with DiT-XL (Peebles & Xie, 2022) takes 16.5 seconds with 100 denoising steps, which is $\sim 10\times$ slower than its smaller counterpart DiT-S (1.7s) with a lower generation quality.

Recent works tackle the inference efficiency issue by speeding up the sampling of DPMs in two ways: (1) reducing the computational costs per step or (2) reducing the number of sampling steps. The former approach can be done by model compression through quantization (Li et al., 2023) and pruning (Fang et al., 2023), or by redesigning lightweight model architectures (Yang et al., 2023; Lee et al., 2023). The second approach reduces the number of steps either by distilling multiple denoising steps into fewer ones (Salimans & Ho, 2022; Song et al., 2023; Zheng et al., 2023) or by improving the differential equation solver (Song et al., 2021a; Lu et al., 2022; Zheng et al., 2023). While both directions can improve the efficiency of large DPMs, they assume that the computational cost of each denoising step remains the same, and a single model is used throughout the process. *However,*

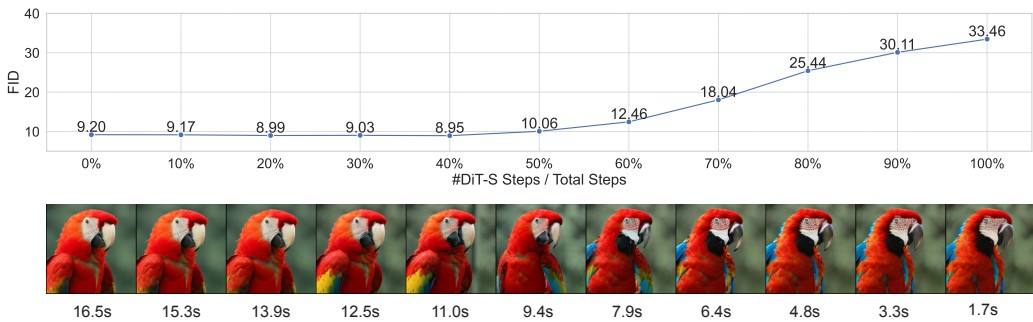

Figure 1: **Top:** FID comparison on class-conditional ImageNet when progressively stitching more DiT-S steps at the beginning and fewer DiT-XL steps in the end, based on DDIM 100 timesteps and a classifier-free guidance scale of 1.5. FID is calculated by sampling 5000 images. **Bottom:** One example of stitching more DiT-S steps to achieve faster sampling, where the time cost is measured by generating 8 images on one RTX 3090 in seconds (s).

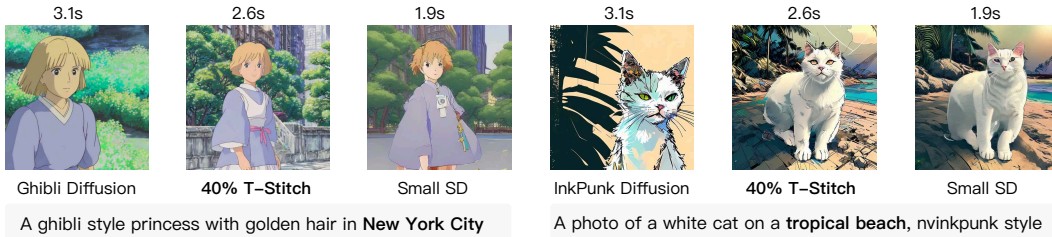

Figure 2: By directly adopting a small SD in the model zoo, T-Stitch naturally interpolates the speed, style, and image contents with a large styled SD, which also potentially improves the prompt alignment, *e.g.*, "New York City" and "tropical beach" in the above examples.

*we observe that different steps in the denoising process exhibit quite distinct characteristics, and using the same model throughout is a suboptimal strategy for efficiency.*

**Our Approach.** In this work, we propose *Trajectory Stitching* (**T-Stitch**), a simple yet effective strategy to improve DPMs' efficiency that complements existing efficient sampling methods by dynamically allocating computation to different denoising steps. *Our core idea is to apply DPMs of different sizes at different denoising steps instead of using the same model at all steps, as in previous works.* We show that by first applying a smaller DPM in the early denoising steps followed by switching to a larger DPM in the later denoising steps, we can reduce the overall computational costs *without* sacrificing the generation quality. Figure 1 shows an example of our approach using two DiT models (DiT-S and DiT-XL), where DiT-S is computationally much cheaper than DiT-XL. With the increase in the percentage of steps from DiT-S instead of DiT-XL in our T-stitch, we can keep increasing the inference speed. In our experiments, we find that there is no degradation of the generation quality (in FID), even when the first 40% of steps are using DiT-S, leading to around 1.5× *lossless* speedup.

Our method is based on two key insights: (1) Recent work suggests a common latent space across different DPMs trained on the same data distribution (Song et al., 2021b; Roeder et al., 2021). Thus, different DPMs tend to share similar sampling trajectories, which makes it possible to stitch across different model sizes and even architectures. (2) From the frequency perspective, the denoising process focuses on generating low-frequency components at the early steps while the later steps target the high-frequency signals (Yang et al., 2023). Although the small models are not as effective for high-frequency details, they can still generate a good global structure at the beginning.

With comprehensive experiments, we demonstrate that T-Stitch substantially speeds up large DPMs without much loss of generation quality. This observation is consistent across a spectrum of architectures and diffusion model samplers. This also implies that T-Stitch can be directly applied to widely used large DPMs without any re-training (*e.g.*, Stable Diffusion (SD) (Rombach et al., 2022)). Figure 2 shows the results of speeding up stylized Stable Diffusion with a relatively smaller pretrained SD model (Kim et al., 2023). Surprisingly, we find that T-Stitch not only improves speed

but also *improves prompt alignment* for stylized models. This is possibly because the fine-tuning process of stylized models (*e.g.*, ghibli, inkpunk) degrades their prompt alignment. T-Stitch improves both efficiency and generation quality here by combining small SD models to complement the prompt alignment for large SD models specialized in stylizing the image.

Note that T-Stitch is *complementary* to existing fast sampling approaches. The part of the trajectory that is taken by the large DPM can still be sped up by reducing the number of steps taken by it, or by reducing its computational cost with compression techniques. In addition, while T-Stitch can already effectively improve the quality-efficiency trade-offs without any overhead of re-training, we show that the generation quality of T-Stitch can be further improved when we fine-tune the stitched DPMs given a trajectory schedule (Section A.12). By fine-tuning the large DPM only on the timesteps that it is applied, the large DPM can better specialize in providing high-frequency details and further improve generation quality. Furthermore, we show that the training-free Pareto frontier generated by T-Stitch improves quality-efficiency trade-offs to training-based methods designed for interpolating between neural network models via model stitching (Pan et al., 2023a;b). Note that T-Stitch is not limited to only two model sizes, and is also applicable to different DPM architectures.

We summarize our main contributions as follows:

- We propose T-Stitch, a simple yet highly effective approach for improving the inference speed of DPMs, by applying a small DPM at early denoising steps while a large DPM at later steps. Without retraining, we achieve better speed and quality trade-offs than individual large DPMs and even non-trivial lossless speedups.
- We conduct extensive experiments to demonstrate that our method is generally applicable to different model architectures and samplers, and is complementary to existing fast sampling techniques.
- Notably, without any re-training overhead, T-Stitch not only accelerates Stable Diffusion models that are widely used in practical applications but also improves the prompt alignment of stylized SD models for text-to-image generation.

## 2 RELATED WORKS

**Efficient diffusion models.** Despite the success, DPMs suffer from the slow sampling speed (Rombach et al., 2022; Ho et al., 2020) due to hundreds of timesteps and the large denoiser (*e.g.*, U-Net). To expedite the sampling process, some efforts have been made by directly utilizing network compression techniques to diffusion models, such as pruning (Fang et al., 2023) and quantization (Shang et al., 2023; Li et al., 2023). On the other hand, many works seek for reducing sampling steps, which can be achieved by distillation (Salimans & Ho, 2022; Zheng et al., 2023; Song et al., 2023), implicit sampler (Song et al., 2021a), improved differential equation (DE) solvers (Lu et al., 2022; Song et al., 2021b; Jolicoeur-Martineau et al., 2021; Liu et al., 2022). Another line of work also considers accelerating sampling by parallel sampling. For example, Zheng et al. (2023) proposed to utilize operator learning to simultaneously predict all steps. Shih et al. (2023) proposed ParaDiGMS to compute the drift at multiple timesteps in parallel. As a complementary technique to the above methods, our proposed trajectory stitching accelerates large DPM sampling by leveraging pretrained small DPMs at early denoising steps, while leaving sufficient space for large DPMs at later steps.

**Multiple experts in diffusion models.** Previous observations have revealed that the synthesis behavior in DPMs can change at different timesteps (Balaji et al., 2022; Yang et al., 2023), which has inspired some works to propose an ensemble of experts at different timesteps for better performance. For example, Balaji et al. (2022) trained an ensemble of expert denoisers at different denoising intervals. However, allocating multiple large denoisers linearly increases the model parameters and does not reduce the computational cost. Yang et al. (2023) proposed a lite latent diffusion model (*i.e.*, LDM) which incorporates a gating mechanism for the wavelet transform in the denoiser to control the frequency dynamics at different steps, which can be regarded as an ensemble of frequency experts. Following the same spirit, Lee et al. (2023) allocated different small denoisers at different denoising intervals to specialize on their respective frequency ranges. Nevertheless, most existing works adopt the same-sized model over all timesteps, which barely consider the speed and quality trade-offs between different-sized models. In contrast, we explore a flexible trade-off between small and large DPMs and reveal that the early denoising steps can be sufficiently handled by a much efficient small DPM.

**Stitchable neural networks.** Stitchable neural networks (SN-Net) (Pan et al., 2023a) is motivated by the idea of model stitching (Lenc & Vedaldi, 2015; Bansal et al., 2021; Csiszárik et al., 2021; Yang et al., 2022), where the pretrained models of different scales within a pretrained model family can be splitted and stitched together with simple stitching layers (*i.e.*, $1 \times 1$ convs) without a significant performance drop. Based on the insight, SN-Net inserts a few stitching layers among models of different sizes and applies joint training to obtain numerous networks (*i.e.*, stitches) with different speed-performance trade-offs. The following work of SN-Netv2 (Pan et al., 2023b) enlarges its space and demonstrates its effectiveness on downstream dense prediction tasks. In this work, we compare our technique with SN-Netv2 to show the advantage of trajectory stitching over model stitching in terms of the speed and quality trade-offs in DPMs. Our T-Stitch is a better, simpler and more general solution.

## 3 METHOD

### 3.1 PRELIMINARY

**Diffusion models.** We consider the class of score-based diffusion models in a continuous time (Song et al., 2021b) and following the presentation from Karras et al. (2022). Let $p_{data}(\mathbf{x}_0)$ denote the data distribution and $\sigma(t)\colon [0,1] \to \mathbb{R}_+$ is a user-specified noise level schedule, where $t \in \{0, ..., T\}$ and $\sigma(t-1) < \sigma(t)$. Let $p(\mathbf{x}; \sigma)$ denote the distribution of noised samples by injecting $\sigma^2$-variance Gaussian noise. Starting with a high-variance Gaussian noise $\mathbf{x}_T$, diffusion models gradually denoise $\mathbf{x}_T$ into less noisy samples $\{\mathbf{x}_{T-1}, \mathbf{x}_{T-2}, ..., \mathbf{x}_0\}$, where $\mathbf{x}_t \sim p(\mathbf{x}_t; \sigma(t))$. Furthermore, this iterative process can be done by solving the probability flow ordinary differential equation (ODE) if knowing the score $\nabla_x \log p_t(x)$, namely the gradient of the log probability density with respect to data,

$$d\mathbf{x} = -\hat{\sigma}(t)\sigma(t)\nabla_{\mathbf{x}} \log p(\mathbf{x}; \sigma(t))\, dt, \qquad (1)$$

where $\hat{\sigma}(t)$ denote the time derivative of $\sigma(t)$. Essentially, diffusion models aim to learn a model for the score function, which can be reparameterized as

$$\nabla_{\mathbf{x}} \log p_t(\mathbf{x}) \approx (D_\theta(\mathbf{x}; \sigma) - \mathbf{x})/\sigma^2, \qquad (2)$$

where $D_\theta(\mathbf{x}; \sigma)$ is the learnable denoiser. Given a noisy data point $\mathbf{x}_0 + \mathbf{n}$ and a conditioning signal $\mathbf{c}$, where $\boldsymbol{n} \sim \mathcal{N}\left(\mathbf{0}, \sigma^2 \boldsymbol{I}\right)$, the denoiser aim to predict the clean data $\mathbf{x}_0$. In practice, the mode is trained by minimizing the loss of denoising score matching,

$$\mathbb{E}_{(\mathbf{x}_0, \mathbf{c}) \sim p_{\text{data}}(\mathbf{x}_0, \mathbf{c}), (\sigma, \mathbf{n}) \sim p(\sigma, \mathbf{n})} \left[ \lambda_\sigma \| D_{\boldsymbol{\theta}}(\mathbf{x}_0 + \mathbf{n}; \sigma, \mathbf{c}) - \mathbf{x}_0 \|_2^2 \right], \qquad (3)$$

where $\lambda_\sigma \colon \mathbb{R}_+ \to \mathbb{R}_+$ is a weighting function (Ho et al., 2020), $p(\sigma, \mathbf{n}) = p(\sigma)\mathcal{N}\left(\mathbf{n}; \mathbf{0}, \sigma^2\right)$, and $p(\sigma)$ is a distribution over noise levels $\sigma$.

This work focuses on *the denoisers* $D$ in diffusion models. In common practice, they are typically large parameterized neural networks with different architectures that consume high FLOPs at each timestep. In the following, we use "denoiser" or "model" interchangeably to refer to this network. We begin with the pretrained DiT model family to explore the advantage of trajectory stitching on efficiency gain. Then we show our method is a general technique for other architectures, such as U-Net (Rombach et al., 2022) and U-ViT (Bao et al., 2023).

**Classifier-free guidance.** Unlike classifier-based denoisers (Dhariwal & Nichol, 2021) that require an additional network to provide conditioning guidance, classifier-free guidance (Ho & Salimans, 2022) is a technique that jointly trains a conditional model and an unconditional model in one network by replacing the conditioning signal with a null embedding. During sample generation, it adopts a guidance scale $s \geq 0$ to guide the sample to be more aligned with the conditioning signal by jointly considering the predictions from both conditional and unconditional models,

$$D^s(\mathbf{x}; \sigma, \mathbf{c}) = (1+s)D(\mathbf{x}; \sigma, \mathbf{c}) - sD(\mathbf{x}; \sigma). \qquad (4)$$

Recent works have demonstrated that classifier-free guidance provides a clear improvement in generation quality. In this work, we consider the diffusion models that are trained with classifier-free guidance due to their popularity.

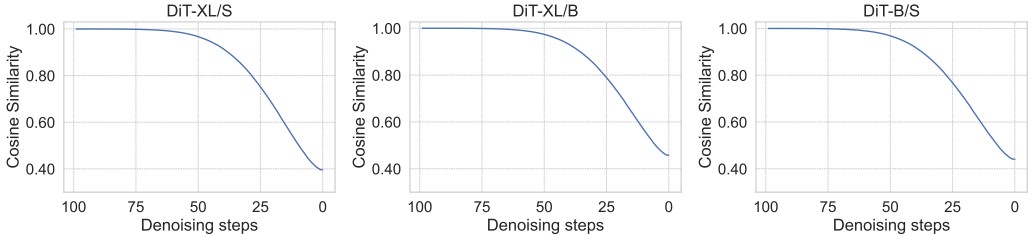

Figure 3: Similarity comparison of latent embeddings at different denoising steps between different DiT models. Results are averaged over 32 images.

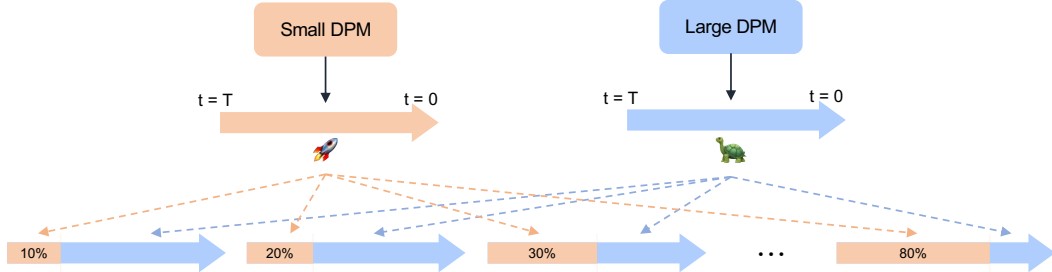

Figure 4: Framework of **Trajectory Stitching** (T-Stitch): Based on pretrained small and large DPMs, we can leverage the more efficient small DPM with different percentages at the early denoising sampling steps to achieve different speed-quality trade-offs.

## 3.2 TRAJECTORY STITCHING

**Why can different pretrained DPMs be directly stitched along the sampling trajectory?** First of all, DPMs from the same model family usually takes the latent noise inputs and outputs of the same shape, (*e.g.*, $4 \times 32 \times 32$ in DiTs). There is no dimension mismatch when applying different DPMs at different denoising steps. More importantly, as pointed out in (Song et al., 2021b), different DPMs that are trained on the same dataset often learn similar latent embeddings. We observe that this is especially true for the latent noises at early denoising sampling steps, as shown in Figure 3, where the cosine similarities between the output latent noises from different DiT models reach almost 100% at early steps. This motivates us to propose Trajectory Stitching (T-Stitch), a novel step-level stitching strategy that leverages a pretrained small model at the beginning to accelerate the sampling speed of large diffusion models.

**Principle of model selection.** Figure 4 shows the framework of our proposed T-Stitch for different speed-quality tradeoffs. In principle, the fast speed or worst generation quality we can achieve is roughly bounded by the smallest model in the trajectory, whereas the slowest speed or best generation quality is determined by the largest denoiser. Thus, given a large diffusion model that we want to speed up, we select a small model that is 1) clearly faster, 2) sufficiently optimized, and 3) trained on the same dataset as the large model or at least they have learned similar data distributions (*e.g.*, pretrained or finetuned stable diffusion models).

**Pairwise model allocation.** By default, T-Stitch adopts a pairwise denoisers in the sampling trajectory as it performs very well in practice. Specifically, we first define a denoising interval as a range of sampling steps in the trajectory, and the fraction of it over the total number of steps $T$ is denoted as $r$, where $r \in [0, 1]$. Next, we treat the model allocation as a compute budget allocation problem. From Figure 3, we observe that the latent similarity between different scaled denoisers keeps decreasing when $T$ flows to 0. To this end, our allocation strategy adopts a small denoiser as a cheap replacement at the initial intervals then applies the large denoiser at the later intervals. In particular, suppose we have a small denoiser $D_1$ and a large denoiser $D_2$. Then we let $D_1$ take the first $\lfloor r_1 T \rceil$ steps and $D_2$ takes the last $\lfloor r_2 T \rfloor$ steps, where $\lfloor \cdot \rceil$ denotes a rounding operation and $r_2 = 1 - r_1$. By increasing $r_1$, we naturally interpolate the compute budget between the small and large denoiser and thus obtain flexible quality and efficiency trade-offs. For example, in Figure 1, the configuration $r_1 = 0.5$ uniquely defines a trade-off where it achieves 10.06 FID and $1.76 \times$ speedup.

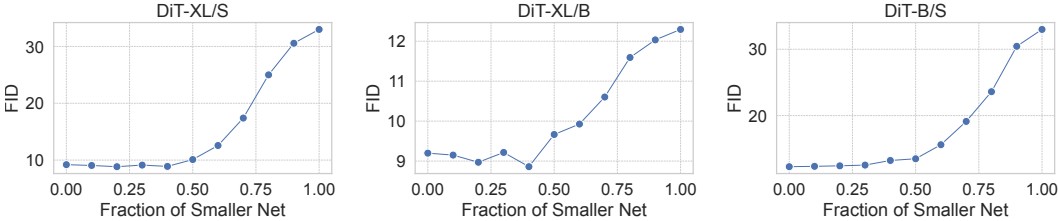

Figure 5: T-Stitch of two model combinations: DiT-XL/S, DiT-XL/B and DiT-B/S. We adopt DDIM 100 timesteps with a classifier-free guidance scale of 1.5.

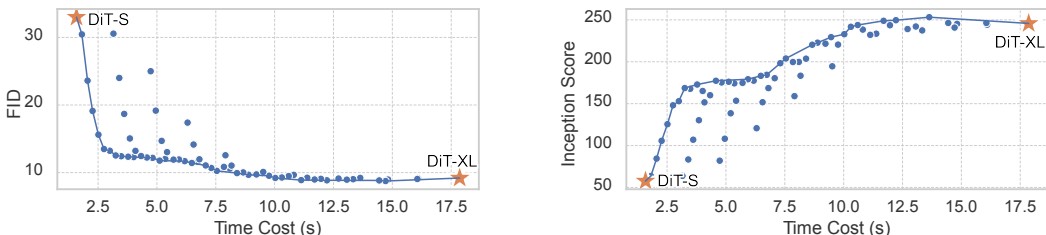

Figure 6: T-Stitch based on three models: DiT-S, DiT-B and DiT-XL. We adopt DDIM 100 timesteps with a classifier-free guidance scale of 1.5. We highlight the Pareto frontier in lines.

**More denoisers for more trade-offs.** Note that T-Stitch is not limited to the pairwise setting. In fact, we can adopt more denoisers in the sampling trajectory to obtain more speed and quality trade-offs and a better Pareto frontier. For example, by using a medium sized denoiser in the intermediate interval, we can change the fractions of each denoiser to obtain more configurations. In practice, given a compute budget such as time cost, we can efficiently find a few configurations that satisfy this constraint via a pre-computed lookup table, as discussed in Section A.1.

**Remark.** Compared to existing multi-experts DPMs, T-Stitch directly applies models of *different sizes* in a *pretrained* model family. Thus, given a compute budget, we consider how to allocate different resources across different steps while benefiting from training-free. Furthermore, the recently proposed speculative decoding (Leviathan et al., 2023) shares a similar motivation with us, *i.e.*, leveraging a small model to speed up large language model sampling. However, this technique is specifically designed for autoregressive models, whereas it is not straightforward to apply the same sampling strategy to diffusion models. On the other hand, our method utilizes the DPM's property and achieves effective speedup.

## 4 EXPERIMENTS

In this section, we first show the effectiveness of T-Stitch based on DiT (Peebles & Xie, 2022) as it provides a convenient model family. Then we extend into U-Net and Stable Diffusion models. Last, we ablate our technique with different sampling steps, and samplers to demonstrate that T-Stitch is generally applicable in many scenarios.

### 4.1 DiT EXPERIMENTS

**Implementation details.** Following DiT, we conduct the class-conditional ImageNet experiments based on pretrained DiT-S/B/XL under 256×256 images and patch size of 2. A detailed comparison of the pretrained models is shown in Table 3. As T-Stitch is training-free, for two-model setting, we directly allocate the models into the sampling trajectory under our allocation strategy described in Section 3.2. For three-model setting, we enumerate all possible configuration sets by increasing the fraction by 0.1 per model one at a time, which eventually gives rise to 66 configurations that include pairwise combinations of DiT-S/XL, DiT-S/B, DiT-S/XL, and three model combinations DiT-S/B/XL. By default, we adopt a classifier-free guidance scale of 1.5 as it achieves the best FID for DiT-XL, which is also the target model in our setting.

**Evaluation metrics.** We adopt Fréchet Inception Distance (FID) (Heusel et al., 2017) as our default metric to measure the overall sample quality as it captures both diversity and fidelity (lower values

Table 1: T-Stitch with LDM (Rombach et al., 2022) and LDM-S on class-conditional ImageNet. All evaluations are based on DDIM and 100 timesteps. We adopt a classifier-free guidance scale of 3.0. The time cost is measured by generating 8 images on one RTX 3090.

| Fraction of LDM-S | 0% | 10% | 20% | 30% | 40% | 50% | 60% | 70% | 80% | 90% | 100% |
|---|---|---|---|---|---|---|---|---|---|---|---|
| FID | 20.11 | 19.54 | 18.74 | 18.64 | 18.60 | 19.33 | 21.81 | 26.03 | 30.41 | 35.24 | 40.92 |
| Inception Score | 199.24 | 201.87 | 202.81 | 204.01 | 193.62 | 175.62 | 140.69 | 110.81 | 90.24 | 70.91 | 54.41 |
| Time Cost (s) | 7.1 | 6.7 | 6.2 | 5.8 | 5.3 | 4.9 | 4.5 | 4.1 | 3.6 | 3.1 | 2.9 |

Table 2: T-Stitch with BK-Tiny (Kim et al., 2023) and SD v1.4. We report FID, Inception Score (IS) and CLIP score (Hessel et al., 2021) on MS-COCO 256×256 benchmark. The time cost is measured by generating one image on one RTX 3090.

| Fraction of BK-Tiny | 0% | 10% | 20% | 30% | 40% | 50% | 60% | 70% | 80% | 90% | 100% |
|---|---|---|---|---|---|---|---|---|---|---|---|
| FID | 13.07 | 12.59 | 12.29 | 12.54 | 13.65 | 14.98 | 15.69 | 16.57 | 16.92 | 16.80 | 17.15 |
| Inception Score | 36.72 | 36.12 | 34.66 | 33.32 | 32.48 | 31.72 | 31.48 | 30.83 | 30.53 | 30.48 | 30.00 |
| CLIP Score | 0.2957 | 0.2957 | 0.2938 | 0.2910 | 0.2860 | 0.2805 | 0.2770 | 0.2718 | 0.2692 | 0.2682 | 0.2653 |
| Time Cost (s) | 3.1 | 3.0 | 2.9 | 2.8 | 2.6 | 2.5 | 2.4 | 2.3 | 2.1 | 2.0 | 1.9 |

indicate better results). Additionally, we report the Inception Score as it remains a solid performance measure on ImageNet, where the backbone Inception network (Szegedy et al., 2016) is pretrained. We use the reference batch from ADM (Dhariwal & Nichol, 2021) and sample 5,000 images to compute FID. In the supplementary material, we show that sampling more images (*e.g.*, 50K) does not affect our observation. By default, the time cost is measured by generating 8 images on a single RTX 3090 in seconds.

**Results.** Based on the pretrained model families, we first apply T-Stitch with any two-model combinations, including DiT-XL/S, DiT-XL/B, and DiT-B/S. For each setting, we begin the sampling steps with a relatively smaller model and then let the larger model deal with the last timesteps. In Figure 5, we report the FID comparisons on different combinations. In general, we observe that using a smaller model at the early 40-50% steps brings a minor performance drop for all combinations. Besides, the best/worst performance is roughly bounded by the smallest and largest models in the pretrained model family.

Furthermore, we show that T-Stitch can adopt a medium-sized model at the intermediate denoising intervals to achieve more speed and quality trade-offs. For example, built upon the three different-sized DiT models: DiT-S, DiT-B, DiT-XL, we start with DiT-S at the beginning then use DiT-B at the intermediate denoising intervals, and finally adopt DiT-XL to draw fine local details. Figure 6 indicates that the three-model combinations effectively obtain a smooth Pareto Frontier for both FID and Inception Score. In particular, at the time cost of ∼10s, we achieve 1.7× speedups with comparable FID (9.21 vs. 9.19) and Inception Score (243.82 vs. 245.73). We show the effect of using different classifier-free guidance scales in Section A.4.

## 4.2 U-NET EXPERIMENTS

In this section, we show T-Stitch is complementary to the architectural choices of denoisers. We experiment with prevalent U-Net as it is widely adopted in many diffusion models. We adopt the class-conditional ImageNet implementation from the latent diffusion model (LDM) (Rombach et al., 2022). Based on their official implementation, we simply scale down the network channel width from 256 to 64 and the context dimension from 512 to 256. This modification produces a 15.8× smaller LDM-S. A detailed comparison between the two pretrained models is shown in Table 4.

**Results.** We report the results on T-Stitch with U-Net in Table 1. In general, under DDIM and 100 timesteps, we found the first ∼50% steps can be taken by an efficient LDM-S with comparable or even better FID and Inception Scores. At the same time, we observe an approximately linear decrease in time cost when progressively using more LDM-S steps in the trajectory. Overall, the U-Net experiment indicates that our method is applicable to different denoiser architectures. We provide the generated image examples in Section A.16 and also show that T-Stitch can be applied with even different model families in Section A.10.

## 4.3 TEXT-TO-IMAGE STABLE DIFFUSION

Benefiting from the public model zoo on Diffusers (von Platen et al., 2022), we can directly adopt a small SD model to accelerate the sampling speed of any large pretrained or styled SD models

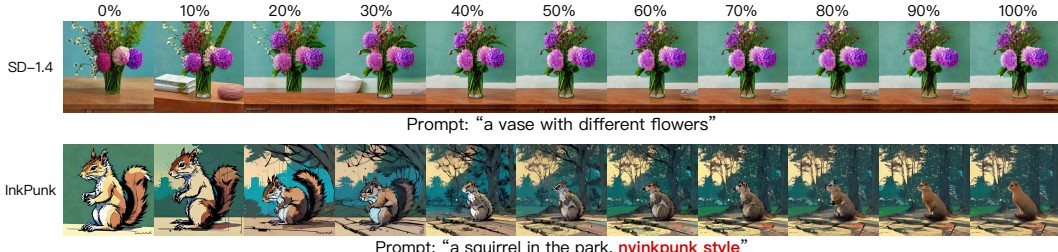

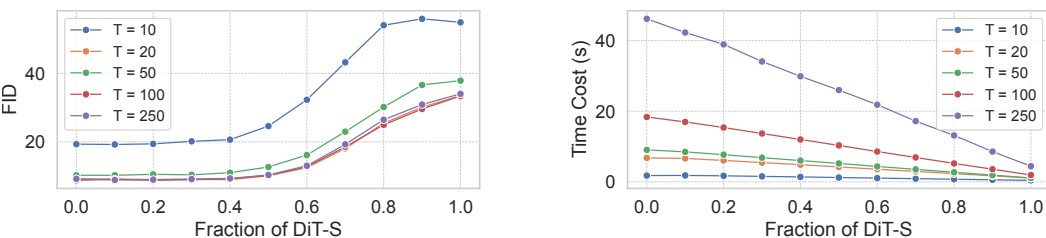

Figure 7: Based on a general pretrained small SD model, T-Stitch simultaneously accelerates a large general SD and complements the prompt alignment with image content when stitching other finetuned/stylized large SD models, *i.e.*, "park" in InkPunk Diffusion. Better viewed when zoomed in digitally.

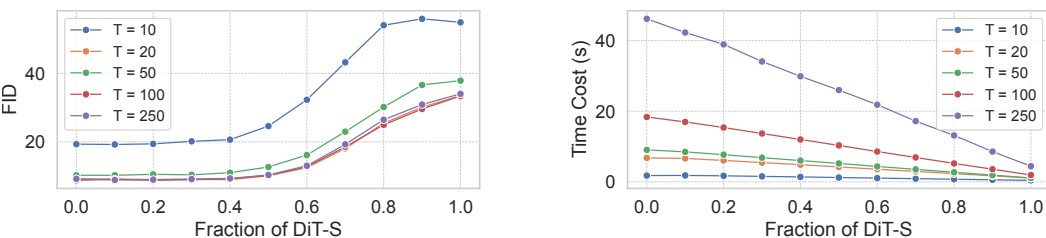

Figure 8: **Left:** we compare FID between different numbers of steps. **Right:** We visualize the time cost of generating 8 images under different number of steps, based on DDIM and a classifier-guidance scale of 1.5. "T" denotes the number of sampling steps.

without any training. In this section, we show how to apply T-Stitch to accelerate existing SD models in the model zoo. Previous research from Kim et al. (2023) has produced multiple SD models with different sizes by pruning the original SD v1.4 and then applying knowledge distillation. We then directly adopt the smallest model BK-SDM Tiny for our stable diffusion experiments. By default, we use a guidance scale of 7.5 under 50 steps using PNDM (Liu et al., 2022) sampler.

**Results.** In Table 2, we report the results by applying T-Stitch to the original SD v1.4. In addition to the FID and Inception Score, we also report the CLIP score for measuring the alignment of the image with the text prompt. Overall, we found the early 30% steps can be taken by BK-SDM Tiny without a significant performance drop in Inception Score and CLIP Scores while achieving even better FID. We believe a better and faster small model in future works can help to achieve better quality and efficiency trade-offs. Furthermore, we demonstrate that T-Stitch is compatible with other large SD models. For example, as shown in Figure 7, under the original SD v1.4, we achieve a promising speedup while obtaining comparable image quality. Moreover, with other stylized SD models such as Inkpunk style[1], we observe a natural style interpolation between the two models. More importantly, by adopting a small fraction of steps from a general small SD, we found it helps the image to be more aligned with the prompt, such as the "park" in InkPunk Diffusion. In this case, we assume finetuning in these stylized SD may unexpectedly hurt prompt alignment, while adopting the knowledge from a general pretrained SD can complement the early global structure generation. Overall, this strongly supports another practical usage of T-Stitch: *Using a small general expert at the beginning for fast sketching and better prompt alignment, while letting any stylized SD at the later steps for patiently illustrating details.* We provide more examples in Section A.11.

### 4.4 ABLATION STUDY

**Effect of T-Stitch with different steps.** To explore the efficiency gain on different numbers of sampling steps, we conduct experiments based on DDIM and DiT-S/XL. As shown in Figure 8, T-Stitch achieves consistent efficiency gain when using different number of steps and diffusion model samplers. In particular, we found the 40% early steps can be safely taken by DiT-S without a significant performance drop. It indicates that small DPMs can sufficiently handle the early denoising steps where they mainly generate the low-frequency components. Thus, we can leave the high-frequency

---

[1]https://huggingface.co/Envvi/Inkpunk-Diffusion

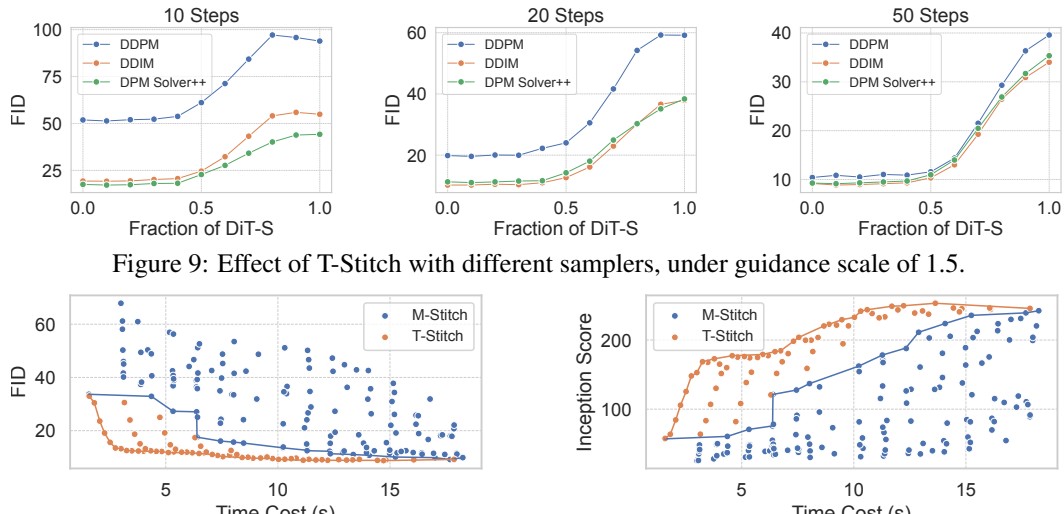

Figure 9: Effect of T-Stitch with different samplers, under guidance scale of 1.5.

Figure 10: T-Stitch (TS) vs. model stitching (MS) (Pan et al., 2023b) based on DiTs and DDIM 100 steps, with a classifier-free guidance scale of 1.5.

generation of fine local details to a more capable DiT-XL. This is further evidenced by the generation examples in Figure 17, where we provide the sampled images at all fractions of DiT-S steps across different total number of steps. In Section A.6, we also show that T-Stitch achieves competitive speed and quality trade-offs with directly reducing sampling steps.

**Effect of T-Stitch with different samplers.** Advanced diffusion model samplers (Lu et al., 2022) have been widely adopted in order to achieve better generation quality in fewer timesteps. To this end, we experiment with prevalent samplers to demonstrate the effectiveness of T-Stitch with these orthogonal techniques: DDPM (Ho et al., 2020), DDIM (Song et al., 2021a) and DPM-Solver++ (Lu et al., 2022). In Figure 9, we use the DiT-S to progressively replace the early steps of DiT-XL under different samplers and steps. In general, we observe a consistent efficiency gain at the initial sampling stage, which strongly justifies that our method is a complementary solution with existing samplers for accelerating DPM sampling.

**T-Stitch vs. model stitching.** Previous works (Pan et al., 2023a;b) such as SN-Net have demonstrated the power of model stitching for obtaining numerous *architectures* that with different complexity and performance trade-offs. Thus, by adopting one of these architectures as the denoiser for sampling, SN-Net naturally achieves flexible quality and efficiency trade-offs. To show the advantage of T-Stitch in the Pareto frontier, we conduct experiments to compare with the framework of model stitching proposed in SN-Netv2 (Pan et al., 2023b). We provide implementation details in Section A.8. In Figure 10, we compare T-Stitch with model stitching based on DDIM sampler and 100 steps. Overall, while both methods can obtain flexible speed and quality trade-offs, T-Stitch achieves clearly better advantage over model stitching across different metrics.

## 5 CONCLUSION

We have proposed Trajectory Stitching, an effective and general approach to accelerate existing pretrained large diffusion model sampling by directly leveraging pretrained smaller counterparts at the initial denoising process, which achieves better speed and quality trade-offs than using an individual large DPM. Comprehensive experiments have demonstrated that T-Stitch achieved consistent efficiency gain across different model architectures, samplers, as well as various stable diffusion models. Besides, our work has revealed the power of small DPMs at the early denoising process. Future work may consider disentangling the sampling trajectory by redesigning or training experts of different sizes at different denoising intervals. For example, designing a better, faster small DPM at the beginning to draw global structures, then specifically optimizing the large DPM at the later stages to refine image details. Besides, more guidelines for the optimal trade-off and more in-depth analysis of the prompt alignment for stylized SDs can be helpful, which we leave for future work.

**Limitations.** T-Stitch requires a smaller model that has been trained on the same data distribution as the large model. Thus, a sufficiently optimized small model is required. Furthermore, adopting an additional small model for denoising sampling will slightly increase memory usage.

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

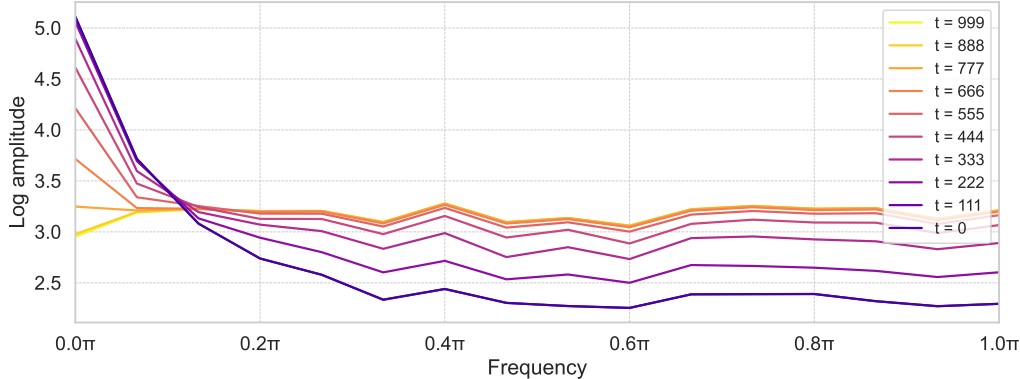

Figure 11: Frequency analysis in denoising process of DiT-XL, based on DDIM 10 steps and guidance scale of 4.0. We visualize the log amplitudes of Fourier-transformed latent noises at each step. Results are averaged over 32 images.

Table 3: Performance comparison of pretrained DiT model family on class-conditional ImageNet. FLOPs are measured by a single forward process given a latent noise in the shape of $4 \times 32 \times 32$.

|  | Parameters (M) | FLOPs (G) | Train Iters | Time Cost (s) | FID |
|---|---|---|---|---|---|
| DiT-S | 33.0 | 5.5 | 5000K | 1.6 | 33.46 |
| DiT-B | 130.5 | 21.8 | 1600K | 4.0 | 12.30 |
| DiT-XL | 675.1 | 114.5 | - | 16.5 | 9.20 |

# A  APPENDIX

## A.1  PRACTICAL DEPLOYMENT OF T-STITCH

In this section, we provide guidelines for the practical deployment of T-Stitch by formulating our model allocation strategy into a compute budget allocation problem.

Given a set of denoisers $\{D_1, D_2, ..., D_K\}$ and their corresponding computational costs $\{C_1, C_2, ..., C_K\}$ for sampling in a $T$-steps trajectory, where $C_{k-1} < C_k$, we aim to find an optimal configuration set $\{r_1, r_2, ..., r_K\}$ that allocates models into corresponding denoising intervals to maximize the generation quality, which can be formulated as

$$\max_{r_1, r_2, ..., r_K} M\left(F(D_1, r_1) \circ F(D_2, r_2) \cdots \circ F(D_K, r_K)\right) \tag{5}$$

$$\text{subject to} \quad \sum_{k=1}^{K} r_k C_k \le C_R, \sum_{k=1}^{K} r_k = 1, \tag{6}$$

where $F(D_k, r_k)$ refers to the denoising process by applying denoiser $D_k$ at the $k$-th interval indicated by $r_k$, $\circ$ denotes to a composition, $M$ represents a metric function for evaluating generation performance, and $C_R$ is the compute budget constraint. Since $\{C_1, C_2, ..., C_K\}$ is known, we can efficiently enumerate all possible fraction combinations and obtain a lookup table, where each fraction configuration set corresponds to a compute budget (*i.e.*, time cost). In practice, we can sample a few configuration sets from this table that satisfy a budget and then apply to generation tasks.

## A.2  FREQUENCY ANALYSIS IN DENOISING PROCESS

We provide evidence that the denoising process focuses on low frequencies at the initial stage and high frequencies in the later steps. Based on DiT-XL, we visualize the log amplitudes of Fourier-transformed latent noises at each sampling step. As shown in Figure 11, the low-frequency amplitudes increase rapidly at the early timesteps (*i.e.*, from 999 to 555), indicating that low frequencies are intensively generated. At the later steps, especially for $t = 111$ and $t = 0$, we observe the

Table 4: Performance comparison of LDM and LDM-S on class-conditional ImageNet.

| Model | Param (M) | Train Iter | Time Cost (s) | FID |
|-------|-----------|------------|---------------|-----|
| LDM-S | 25 | 400K | 2.9 | 40.92 |
| LDM | 394 | 200K | 7.1 | 20.11 |

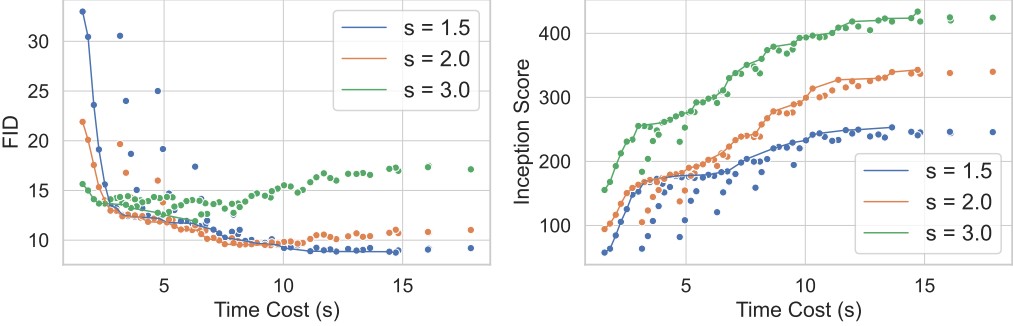

Figure 12: Trajectory stitching based on three models: DiT-S, DiT-B, and DiT-XL. We adopt DDIM 100 timesteps with a classifier-free guidance scale of 1.5, 2.0 and 3.0.

log amplitude of high frequencies increases significantly, which implies that the later steps focus on detail refinement.

### A.3 PRETRAINED DiTS AND U-NETS

In Table 3 and Table 4, we provide detailed comparisons of the pretrained DiT model family, as well as our reproduced small version of U-Net. Overall, as mentioned earlier in Section 3, we make sure the models at each model family have a clear gap in model size between each other such that we can achieve a clear speedup.

### A.4 EFFECT OF DIFFERENT CLASSIFIER-FREE GUIDANCE ON THREE-MODEL TRAJECTORY STITCHING

In Figure 12, we provide the results by applying T-Sittch with DiTs using different guidance scales under three-model settings. In general, T-Stitch performs consistently with different guidance scales, where it interpolates a smooth Pareto frontier between the DiT-S and DiT-XL. As common practice in DPMs adopt different guidance scales to control image generation, this significantly underscores the broad applicability of T-Stitch.

### A.5 FID-50K VS. FID-5K

For efficiency concerns, we report FID based on 5,000 images by default. Based on DiT, we apply T-Stitch with DDPM 250 steps with a guidance scale of 1.5 and sample 50,000 images for evaluating FID. As shown in Figure 13, the observation between FID-50K and FID-5K are similar, which indicates that sampling more images like 50,000 does not affect the effectiveness.

### A.6 COMPARED TO DIRECTLY REDUCING SAMPLING STEPS

Reducing sampling steps has been a common practice for obtaining different speed and quality trade-offs during deployment. Although we have demonstrated that T-Stitch can achieve consistent efficiency gain under different sampling steps, we show in Figure 15 that compared to directly reducing the number of sampling steps, the trade-offs from T-Stitch are very competitive, especially for the 50-100 steps region where the FIDs under T-Stitch are even better. Thus, T-Stitch is able to serve as a complementary or an alternative method for practical DPM sampling speed acceleration.

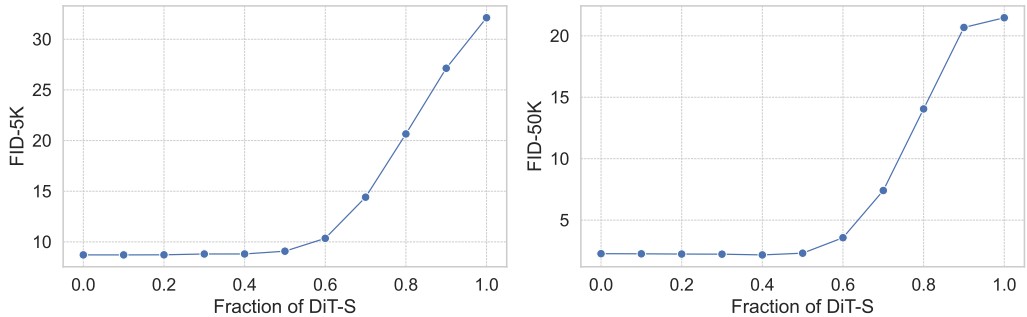

Figure 13: Trajectory stitching based on three models: DiT-S, DiT-B, and DiT-XL. We adopt DDPM 250 timesteps with a classifier-free guidance scale of 1.5.

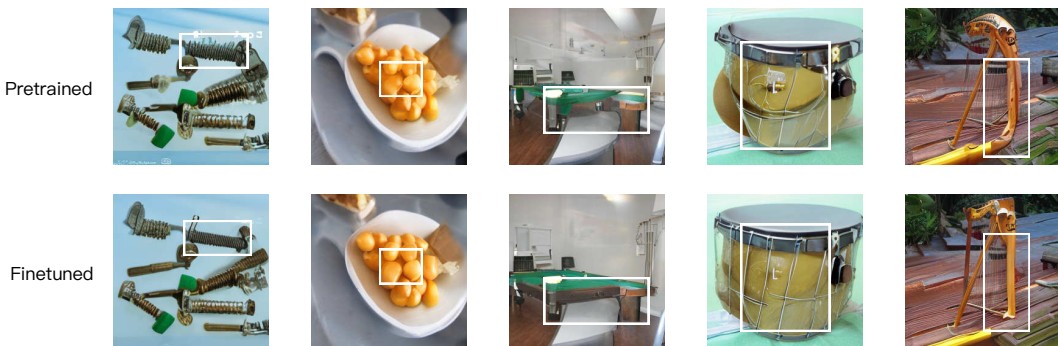

Figure 14: Image quality comparison by stitching pretrained and finetuned DiT-B and DiT-XL at the later steps, based on T-Stitch schedule of DiT-S/B/XL of 50% : 30% : 20%.

### A.7 MODEL COMPRESSION AT THE LATER STEPS

In practice, T-Stitch is orthogonal to individual model optimization. For example, with a BK-SDM Tiny and SDv1.4, we can still apply compression into SDv1.4 in order to reduce the computational cost at the later steps from the large SD. In Figure 16, we show that by adopting a compressed SD v1.4, *i.e.*, BK-SDM Small, we can further reduce the time cost with a trade-off for image quality.

### A.8 IMPLEMENTATION DETAILS OF MODEL STITCHING BASELINE

We adopt a LoRA rank of 64 when stitching DiT-S/XL, which leads to 134 stitching configurations. The stitched model is finetuned on 8 A100 GPUs for 1,700K training iterations. We pre-extract the ImageNet features with a Stable Diffusion AutoEncoder (Rombach et al., 2022) and do not apply any data augmentation. Following the baseline DiT, we adopt the AdamW optimizer with a constant learning rate of $1 \times 10^{-4}$. The total batch size is set as 256. All other hyperparameters adopt the default setting as DiT.

### A.9 IMAGE EXAMPLES UNDER THE DIFFERENT NUMBER OF SAMPLING STEPS

Figure 17 shows image examples generated using different numbers of sampling steps under T-Stitch and DiT-S/XL. As the figure shows, adopting a small model at the early 40% steps has a negligible effect on the final generated images. When progressively increasing the fraction of DiT-S, there is a visible trade-off between speed and quality, with the final image becoming more similar to those generated from DiT-S.

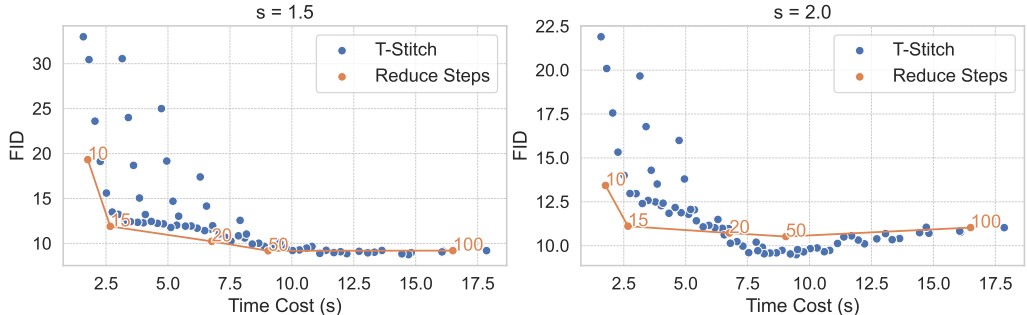

Figure 15: Based on DDIM, we report the FID and speedup comparisons on DiT-XL by using T-Stitch and directly reducing the sampling step from 100 to 10. "s" denotes the classifier-free guidance scale. Trajectory stitching adopts the three-model combination (DiT-S/B/XL) under 100 steps.

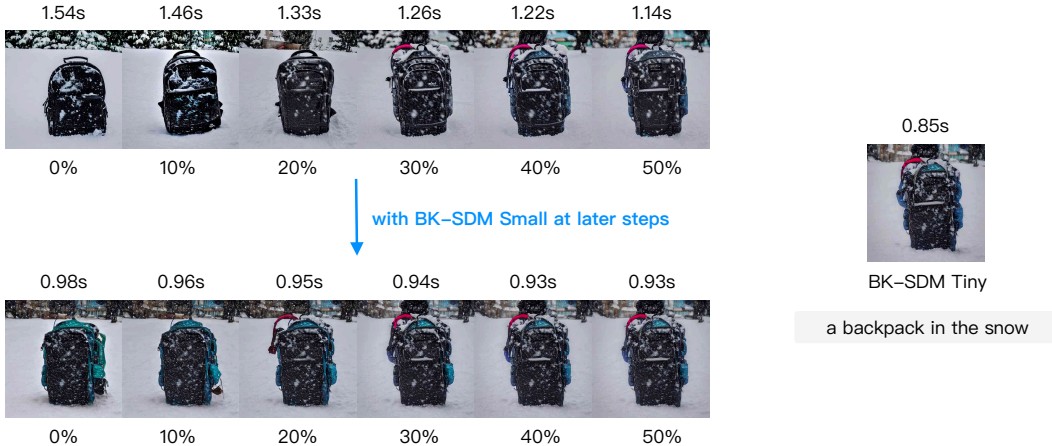

Figure 16: Comparison of T-Stitch by adopting SDv1.4 and its compressed version (*i.e.*, BK-SDM Small) at the later steps.

## A.10 T-STITCH WITH DIFFERENT PRETRAINED MODEL FAMILIES

As different pretrained models trained on the same dataset to learn similar encodings, T-Stitch is able to directly integrate different pretrained model families. For example, based on U-ViT H (Bao et al., 2023), we apply DiT-S at the early sampling steps just as we have done for DiTs and U-Nets. Remarkably, as shown in Table 5, it performs very well, which demonstrates the advantage of T-Stitch as it can be applied for more different models in the public model zoo.

## A.11 MORE EXAMPLES IN STABLE DIFFUSION

We show more examples by applying T-Stitch to SD v1.4, InkPunk Diffusion and Ghibli Diffusion with a small SD model, BK-SDM Tiny (Kim et al., 2023). For all examples, we adopt the default scheduler and hyperparameters of StableDiffusionPipeline in Diffusers: PNDM scheduler, 50 steps, guidance scale 7.5. In Figure 21, we observe that adopting a small SD in the sampling trajectory of SD v1.4 achieves minor effect on image quality at the small fractions and obtain flexible trade-offs in speed and quality by using different fractions. In Figure 24, we show T-Stitch performs favorably with more complex prompts. Besides, by adopting a smaller and distilled SSD-1B, we can easily accelerate SDXL while being compatible with complex prompts and ControlNet (Zhang et al., 2023) for practical art generation, as shown in Figure 25 and Figure 26. Furthermore, we demonstrate that T-Stitch is robust in practical usage. As shown in Figure 27, 8 consecutive runs can generate stable

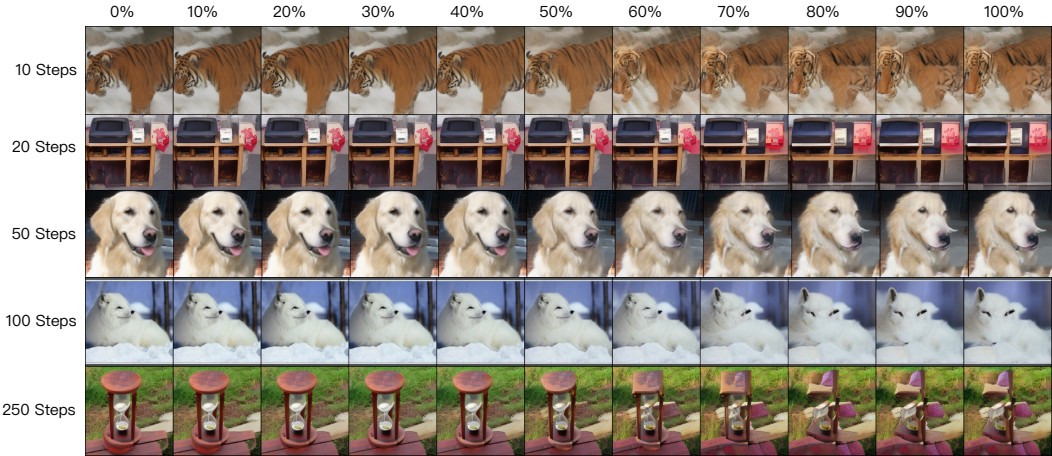

Figure 17: Based on DDIM and a classifier-free guidance scale of 1.5, we stitch the trajectories from DiT-S and DiT-XL and progressively increase the fraction (%) of DiT-S timesteps at the beginning.

Table 5: T-Stitch with DiT-S and U-ViT H, under DPM-Solver++, 50 steps, guidance scale of 1.5.

| Fraction of DiT-S | 0% | 10% | 20% | 30% | 40% | 50% | 60% | 70% | 80% | 90% | 100% |
|---|---|---|---|---|---|---|---|---|---|---|---|
| FID | 15.04 | 13.68 | 12.44 | 12.76 | 14.19 | 17.6 | 29.4 | 53.75 | 74.14 | 84.33 | 121.95 |
| Time Cost (s) | 15.90 | 13.21 | 11.91 | 10.61 | 9.42 | 7.92 | 6.57 | 5.23 | 3.84 | 2.50 | 1.40 |

images with great quality. For stylized SDs, such as InkPunk-Diffusion and Ghibli-Diffusion[2], we show in Figures 22 and 23 that T-Stitch helps to complement the prompt alignment by effectively utilizing the knowledge of the pretrained small SD. Benefiting from the interpolation on speeds, styles and image contents, T-Stitch naturally increases the diversity of the generated images given a prompt by using different fractions of small SD.

## A.12 FINETUNING ON SPECIFIC TRAJECTORY SCHEDULE

When progressively using a small model in the trajectory, we observe a non-negligible performance drop. However, we show that we can simply finetune the model at the allocated denoising intervals to improve the generation quality. For example, based on DDIM and 100 steps, allocating DiT-S at the early 50%, DiT-B at the subsequent 30%, and DiT-XL at the last 20% obtains an FID of 16.49. In this experiment, we separately finetune DiT-B and DiT-XL at their allocated denoising intervals, with additional 250K iterations on ImageNet-1K under the default hyperparameters in DiT (Peebles & Xie, 2022). In Table 6, we observe a clear improvement over FID, Precision and Recall. Furthermore, we provide a comparison of the generated images in Figure 14, where we observe that finetuning clearly improves local details.

Table 6: Performance comparison of stitching pretrained and finetuned DiTs at the later steps. We set the denoising interval of DiT-S/B/XL with 50% : 30% : 20%

| | FID | Inception Score |
|---|---|---|
| Pretrained | 16.49 | 123.11 |
| Finetuned | 13.35 | 155.35 |

## A.13 COMPARED WITH MORE STITCHING BASELINES

By default, we design T-Stitch to start from a small DPM and then switch into a large DPM for the last denoising sampling steps. To show the effectiveness of this design, we compare our method with several baselines in Table 7 based on DiT-S and DiT-XL, including

---

[2]https://huggingface.co/nitrosocke/Ghibli-Diffusion

Table 7: Compared to other trajectory stitching baselines based on DiT-S/XL, DDIM 100 steps and guidance scale of 1.5. FID is calculated by 5K images. Memory and time cost are measured by a batch size of 8 on one RTX 3090.

| Method | FID | Inception Score | Time Cost |
|---|---|---|---|
| Interleave | 19.02 | 120.04 | 10.1 |
| Decreasing Prob | 12.94 | 163.45 | 9.8 |
| Large to Small | 27.61 | 72.60 | 10.0 |
| Small to Large (Ours) | 10.06 | 200.81 | 9.9 |

Table 8: Local storage and memory cost comparison between DiT-S, DiT-XL and T-Stitch. Memory and time cost are measured by generating 8 images in parallel on one RTX 3090.

| Name | Parameter (M) | Local Storage (MB) | Memory Cost (MB) | Time Cost (s) |
|---|---|---|---|---|
| DiT-S | 33 | 263 | 3088 | 1.7 |
| DiT-XL | 675 | 2576 | 3166 | 16.5 |
| T-Stitch (50%) | 708 ($\times 1.04$) | 2839 ($\times 1.10$) | 3296 ($\times 1.04$) | 9.4 ($\times 1.76$) |

- **Interleaving.** During denoising sampling, we interleave the small and large model along the trajectory. Eventually, DiT-S takes 50% steps and DiT-XL takes another 50% steps.

- **Decreasing Prob.** Linearly decreasing the probability of using DiT-S from 1 to 0 during the denoising sampling steps.

- **Large to Small.** Adopting the large model at the early 50% steps and the small model at the last 50% steps.

- **Small to Large (our default design).** The default strategy of T-Stitch by adopting DiT-S at the early 50% steps and using DiT-XL at the last 50% steps.

As shown in Table 7, in general, our default design achieves the best FID and Inception Score with similar sampling speed, which strongly demonstrate its effectiveness.

### A.14 ADDITIONAL MEMORY AND STORAGE OVERHEAD OF T-STITCH

Intuitively, T-Stitch adopts a small DPM which can introduce additional memory and storage overhead. However, in practice, the large DPM is still the main bottleneck of memory and storage consumption. In this case, the additional overhead from small DPM is considerably minor. For example, as shown in Table 8, compared to DiT-XL, T-Stitch by adopting 50% steps of DiT-S only introduces additional 5% parameters, 4% GPU memory cost, 10% local storage cost, while significantly accelerating DiT-XL sampling speed by $1.76\times$.

### A.15 PRECISION AND RECALL MEASUREMENT OF T-STITCH

Following common practice (Dhariwal & Nichol, 2021), we adopt Precision to measure fidelity and Recall to measure diversity or distribution coverage. In Table 9, we show that T-Stitch introduces a minor effect on Precision and Recall at the early 40-50% steps, while at the later steps we observe clear trade-offs, which is consistent with FID evaluations.

### A.16 IMAGE EXAMPLES OF T-STITCH ON DITS AND U-NETS

In Figures 19 and 20, we provide image examples that generated by applying T-Stitch with DiT-S/XL, LDM-S/LDM, respectively. Overall, we observe that adopting a small DPM at the beginning still produces meaningful and high-quality images, while at the later steps it achieves flexible speed and quality trade-offs. Note that different from DiTs that learn a null class embedding during

Table 9: Precision and Recall evaluation based on DiT-S/XL, with DDIM 100 steps and guidance scale of 1.5.

| Fraction of DiT-S | 0% | 10% | 20% | 30% | 40% | 50% | 60% | 70% | 80% | 90% | 100% |
|---|---|---|---|---|---|---|---|---|---|---|---|
| FID | 9.20 | 9.17 | 8.99 | 9.03 | 8.95 | 10.06 | 12.46 | 18.04 | 25.44 | 30.11 | 33.46 |
| Precision | 0.81 | 0.81 | 0.81 | 0.81 | 0.80 | 0.76 | 0.72 | 0.67 | 0.62 | 0.59 | 0.58 |
| Recall | 0.74 | 0.74 | 0.74 | 0.74 | 0.75 | 0.75 | 0.74 | 0.73 | 0.69 | 0.65 | 0.63 |

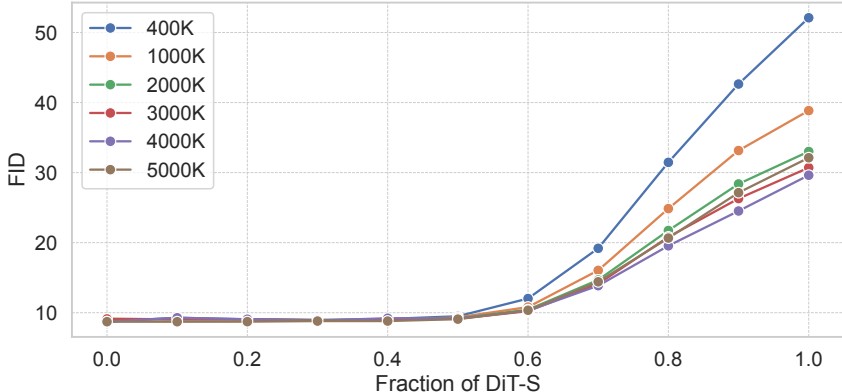

Figure 18: Effect of different pretrained DiT-S in T-Stitch for accelerating DiT-XL, based on DDPM, 250 steps and guidance scale of 1.5. For example, "400K" indicates the pretrained weights of DiT-S at 400K iterations.

classifier-free guidance, LDM inherently omits this embedding in their official implementation [3]. During sampling, LDM and LDM-S have different unconditional signals, which eventually results in various image contents under different fractions.

### A.17 EFFECT OF DIT-S UNDER DIFFERENT TRAINING ITERATIONS

In our experiments, we adopt a DiT-S that trained with 5,000K iterations as it can be sufficiently optimized. In Figure 18, we indicate that even under a short training schedule of 400K iterations, adopting DiT-S at the initial stages of the sampling trajectory also has a minor effect on the overall FID. The main difference is at the later part of the sampling trajectory. Therefore, it implies the early denoising sampling steps can be easier to learn and be handled by a compute-efficient small model.

---

[3]https://github.com/CompVis/latent-diffusion

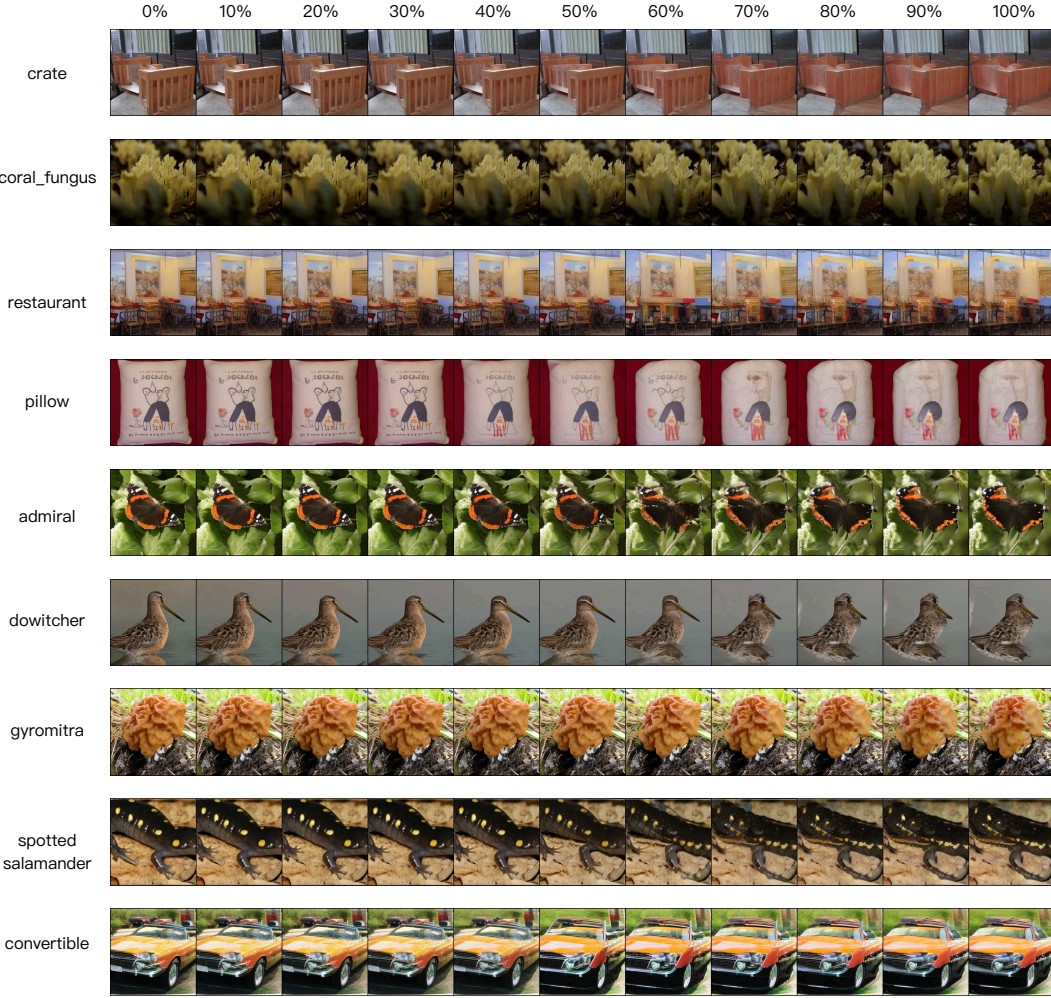

Figure 19: Image examples of T-Stitch on DiT-S and DiT-XL. We adopt DDIM and 100 steps, with a guidance scale of 4.0. From left to right, we gradually increase the fraction of LDM-S steps at the beginning, then let the original LDM to process later denoising steps.

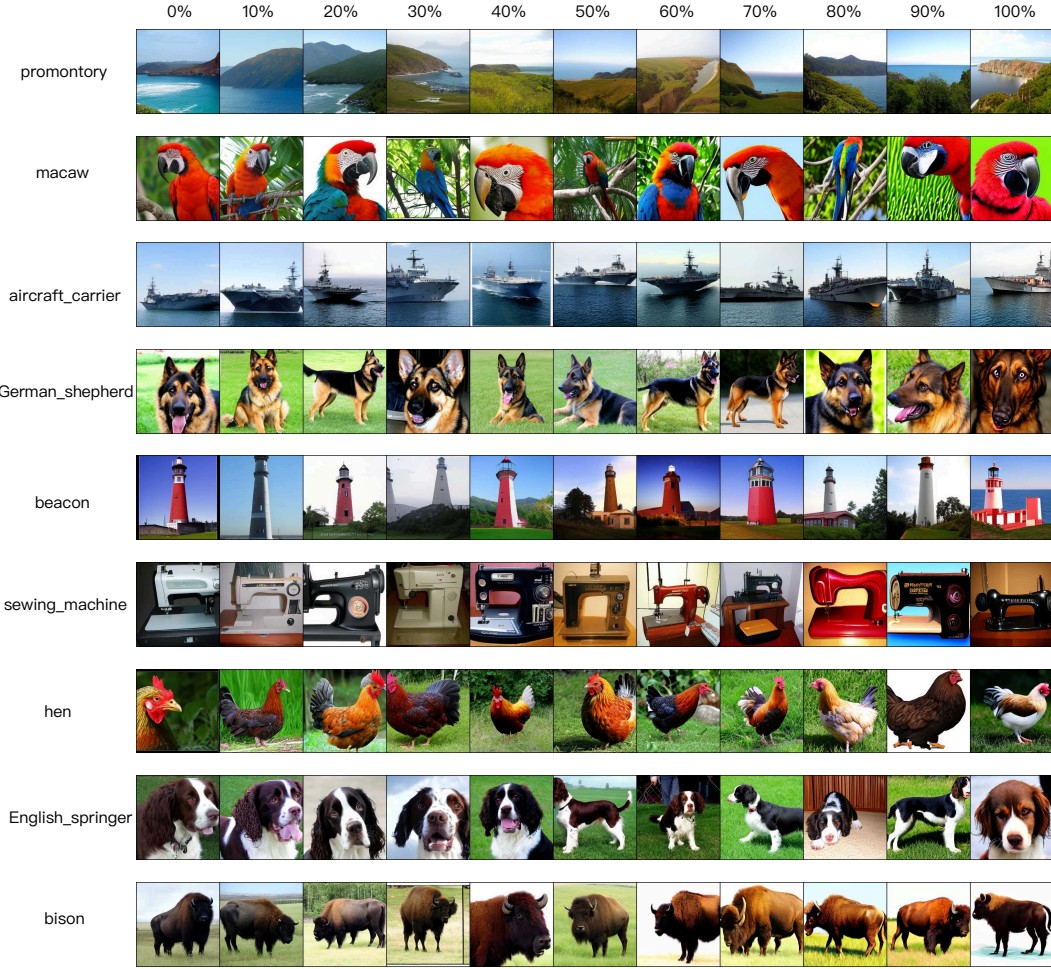

Figure 20: Image examples of T-Stitch on U-Net-based LDM and LDM-S. We adopt DDIM and 100 steps, with a guidance scale of 3.0. From left to right, we gradually increase the fraction of LDM-S steps at the beginning, then let the original LDM to process later denoising steps.

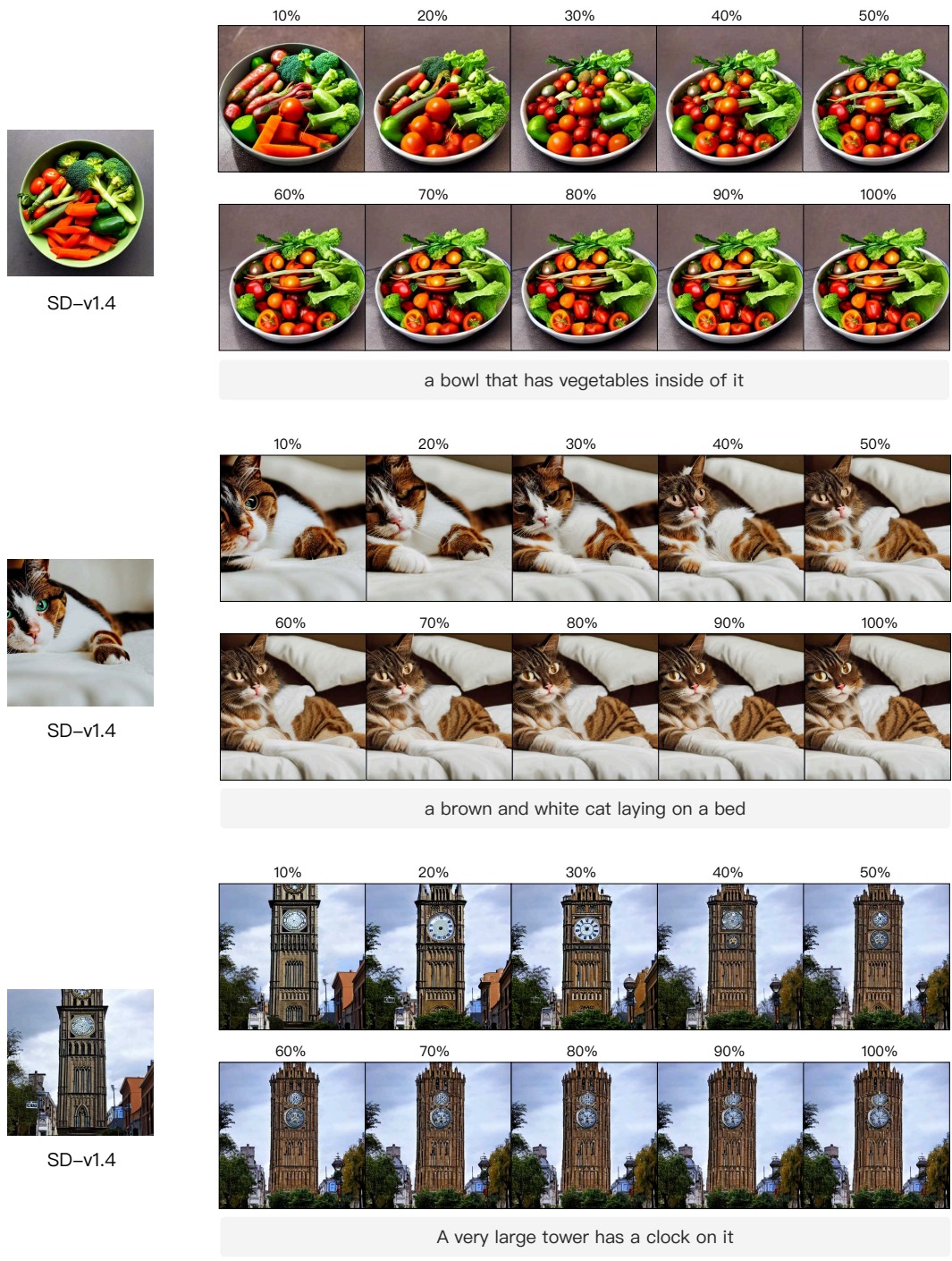

Figure 21: T-Stitch based on Stable Diffusion v1.4 and BK-SDM Tiny. We annotate the faction of BK-SDM on top of images.

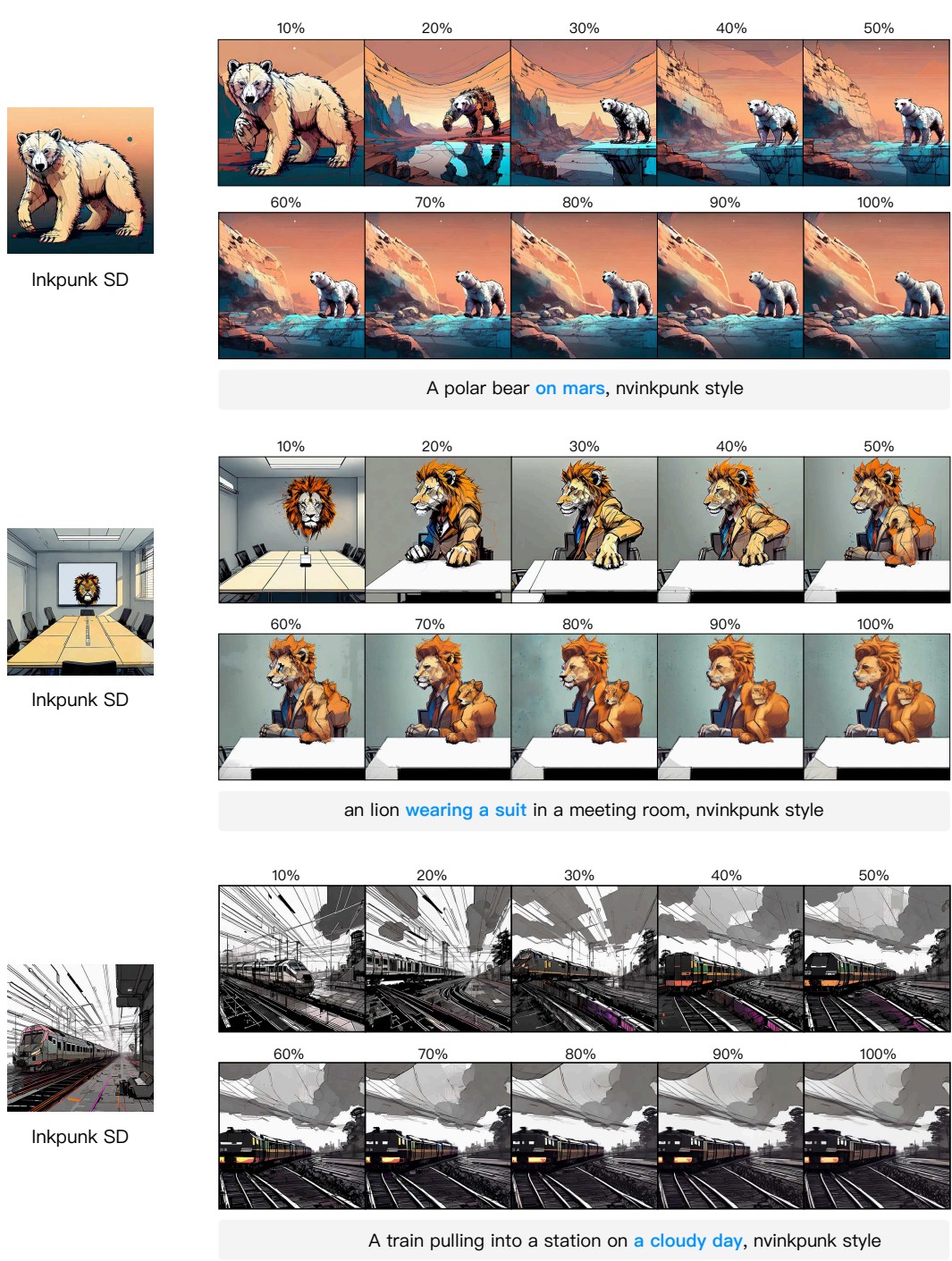

Figure 22: T-Stitch based on Inkpunk-Diffusion SD an BK-SDM Tiny. We annotate the faction of BK-SDM on top of images.

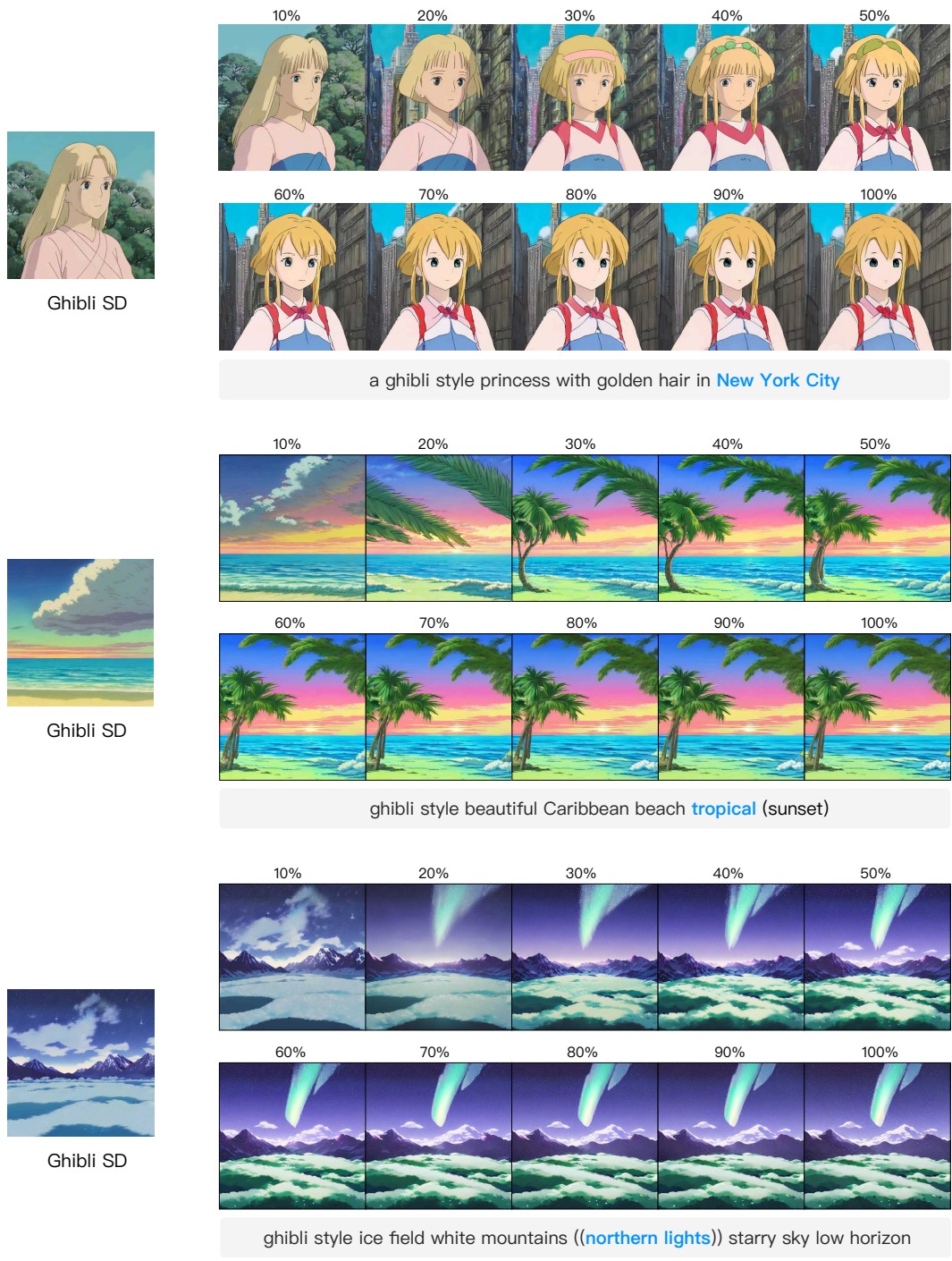

Figure 23: T-Stitch based on Ghibli-Diffusion SD and BK-SDM Tiny. We annotate the faction of BK-SDM on top of images.

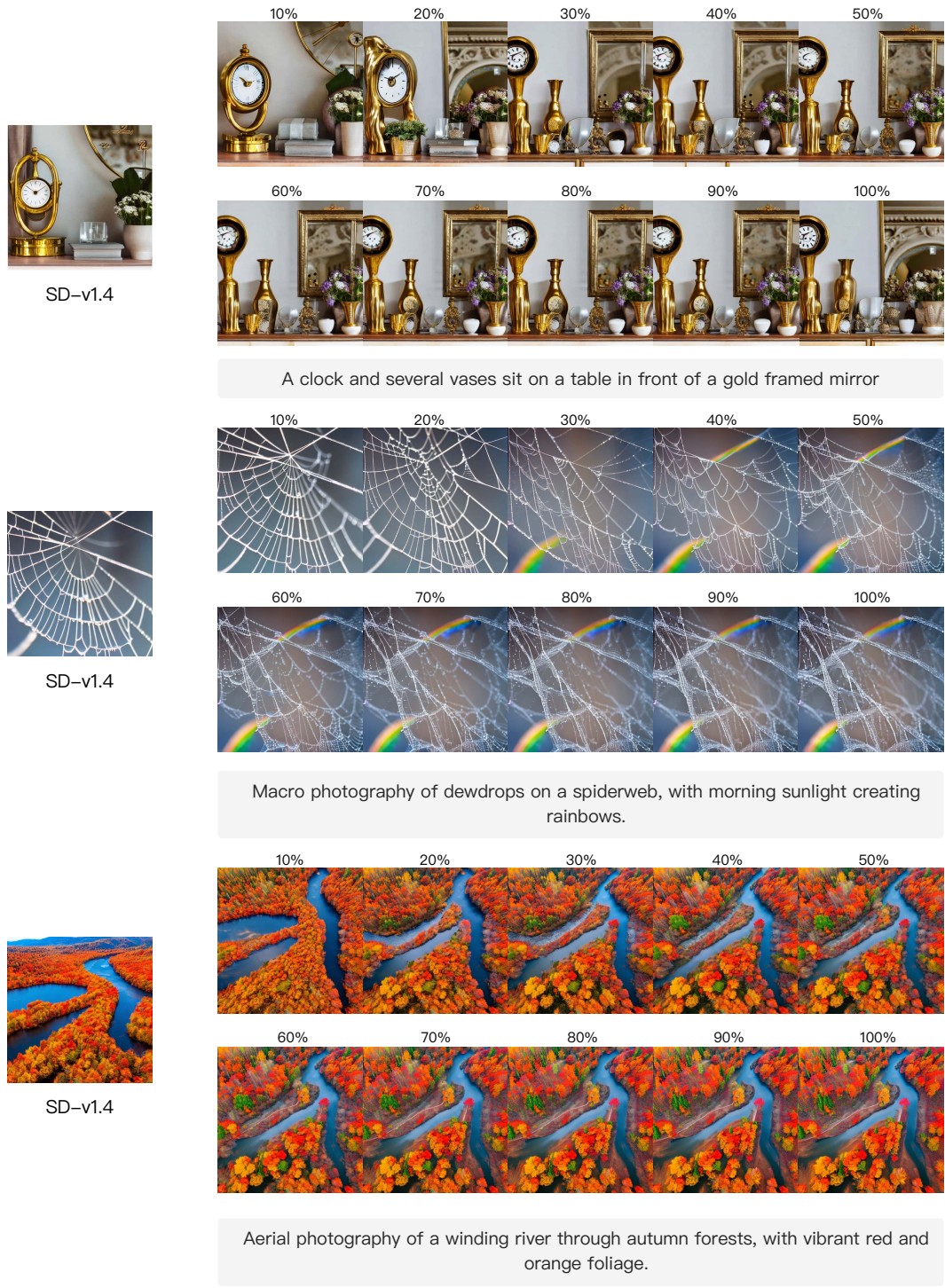

Figure 24: T-Stitch with more complex prompts based on Stable Diffusion v1.4 and BK-SDM Tiny. We annotate the faction of BK-SDM on top of images.

SDXL, 13.6s    20%, 12.04s    40%, 11.05s    60%, 10.6s    80%, 9.7s    100%, 8.8s

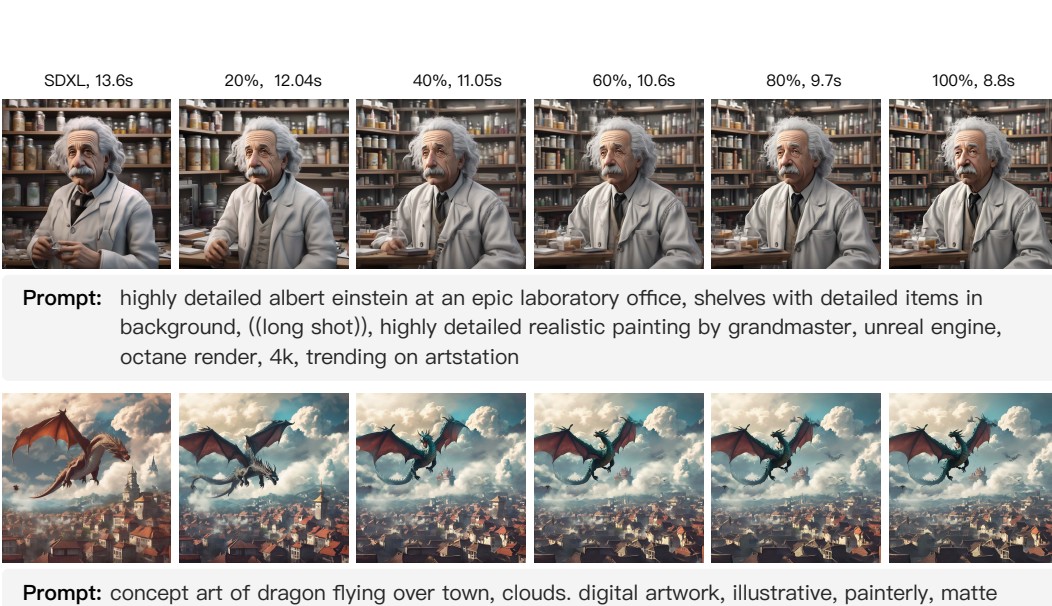

**Prompt:** highly detailed albert einstein at an epic laboratory office, shelves with detailed items in background, ((long shot)), highly detailed realistic painting by grandmaster, unreal engine, octane render, 4k, trending on artstation

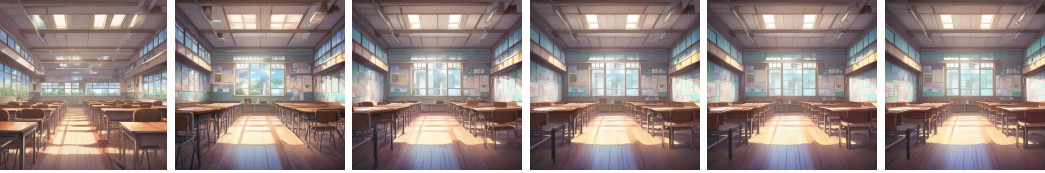

**Prompt:** concept art of dragon flying over town, clouds. digital artwork, illustrative, painterly, matte painting, highly detailed, cinematic composition

**Negative Prompt:** photo, photorealistic, realism, ugly

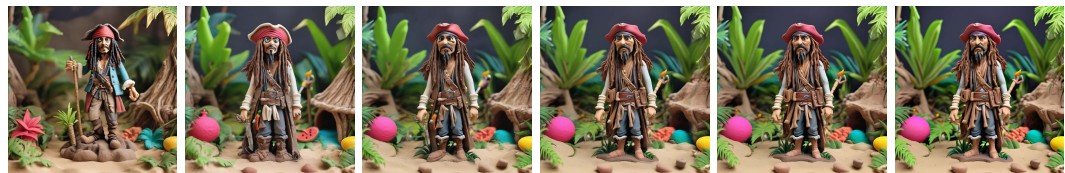

**Prompt:** anime artwork an empty classroom. anime style, key visual, vibrant, studio anime, highly detailed

**Negative Prompt:** photo, deformed, black and white, realism, disfigured, low contrast

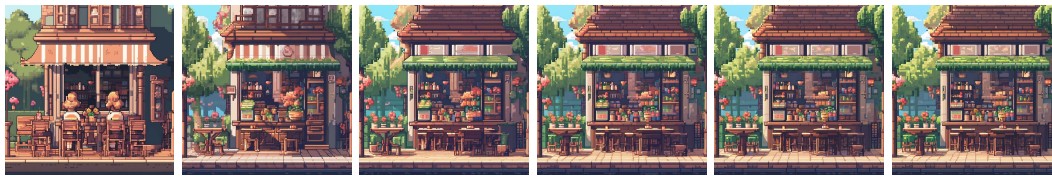

**Prompt:** claymation style captain jack sparrow on tropical island. sculpture, clay art, centered composition, play–doh

**Negative Prompt:** sloppy, messy, grainy, highly detailed, ultra textured, photo, mutated

**Prompt:** 16–bit pixel art, a cozy cafe side view, a beautiful day

**Negative Prompt:** sloppy, messy, blurry, noisy, highly detailed, ultra textured, photo, realistic

Figure 25: T-Stitch with more complex prompts based on SDXL (Podell et al., 2023) and SSD-1B (Segmind, 2023). We annotate the faction of SSD-1B on top of images. Time cost is measured by generating one image on RTX 3090.

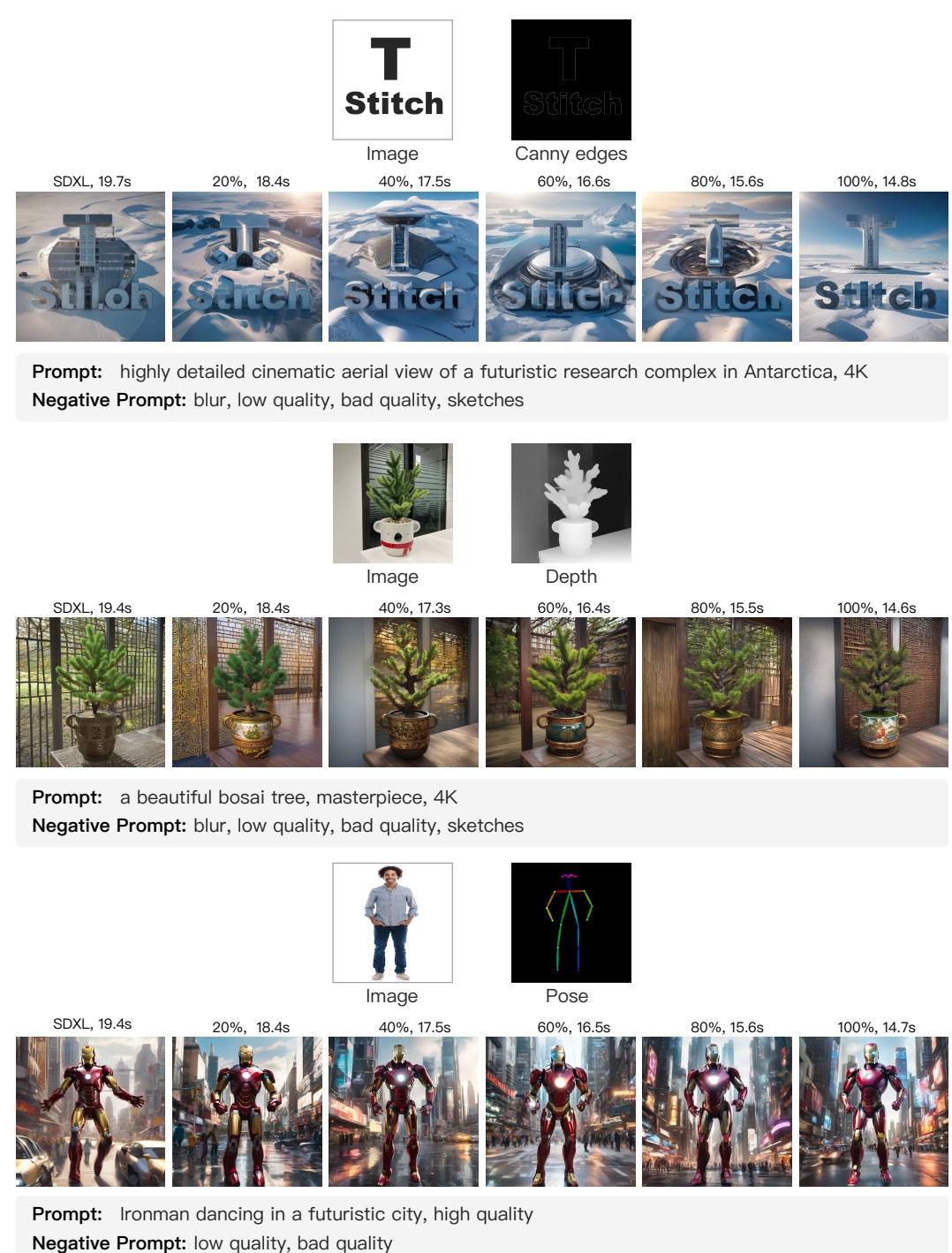

Figure 26: T-Stitch with SDXL-based ControlNet. We annotate the faction of SSD-1B on top of images. Time cost is measured by generating one image on one RTX 3090.

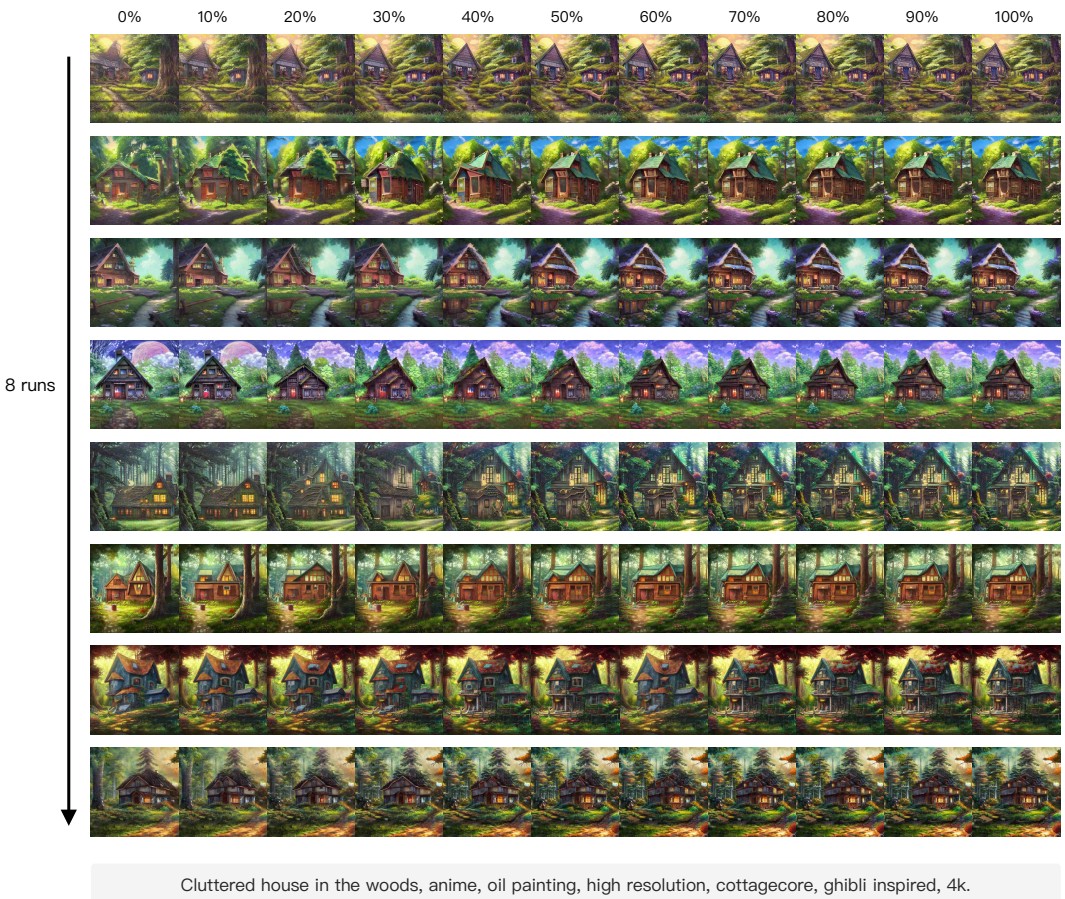

Cluttered house in the woods, anime, oil painting, high resolution, cottagecore, ghibli inspired, 4k.

Figure 27: Based on Stable Diffusion v1.4 and BK-SDM Tiny, we generate images by different fractions of BK-SDM for **8 consecutive runs** (a for-loop) on one GPU. T-Stitch demonstrates stable performance for robust image generation. Best viewed in digital version and zoom in.

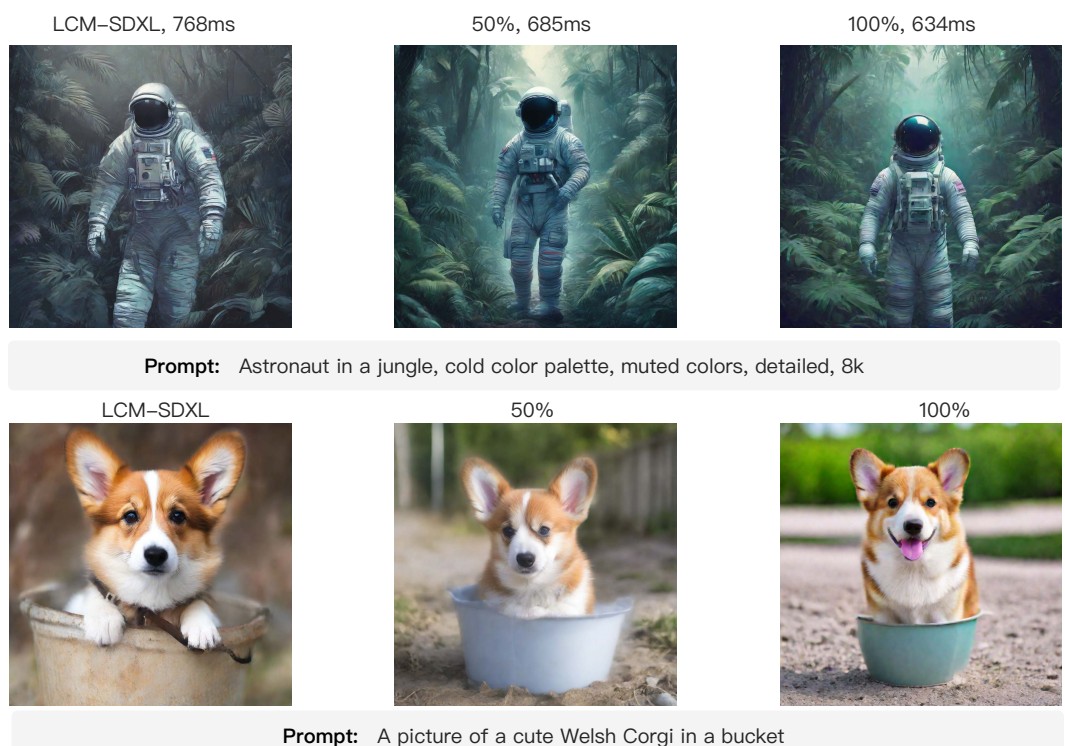

Figure 28: T-Stitch based on distilled models: LCM-SDXL (Luo et al., 2023) and LCM-SSD-1B (Luo et al., 2023), under **2 sampling steps**. We annotate the faction of LCM-SSD-1B on top of images. Time cost is measured by generating one image on RTX 3090 in milliseconds.

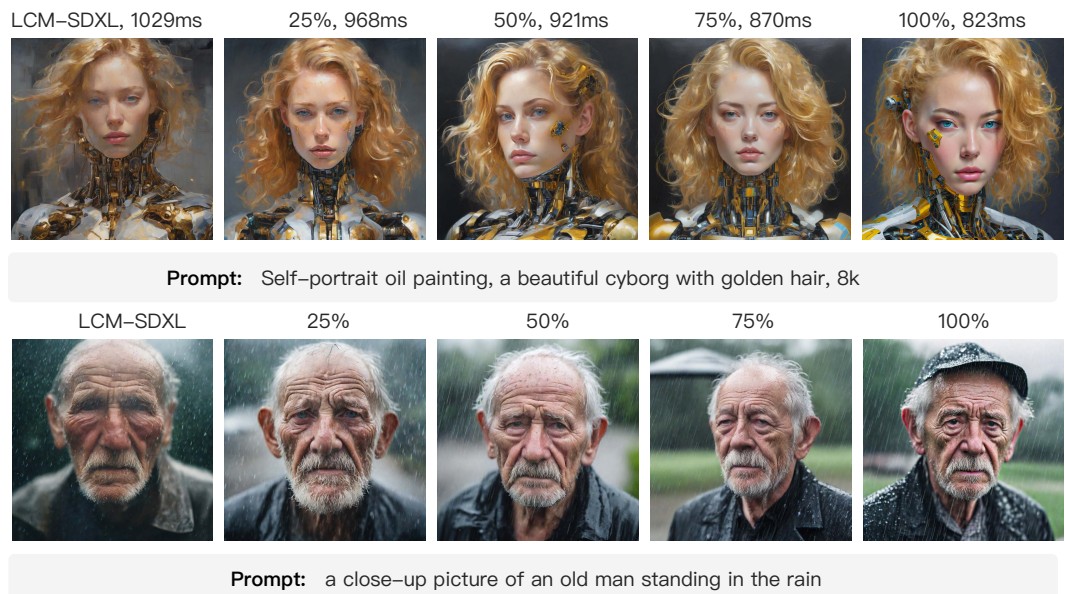

Figure 29: T-Stitch based on distilled models: LCM-SDXL (Luo et al., 2023) and LCM-SSD-1B (Luo et al., 2023), under **4 sampling steps**. We annotate the faction of LCM-SSD-1B on top of images. Time cost is measured by generating one image on RTX 3090 in milliseconds.

