# OpenReview forum: "T-Stitch: Accelerating Sampling in Pre-Trained Diffusion Models with Trajectory Stitching"
_ICLR.cc/2024/Conference — Submitted to ICLR 2024_

### Official Review · Reviewer_UhCD · 2023-10-31

**Soundness:** 2 fair
**Presentation:** 2 fair
**Contribution:** 2 fair
**Rating:** 3
**Confidence:** 4

**Summary:**

This paper proposes a technique called trajectory stitching (T-Stich) to reduce the overall inference time of DPMs while maintaining the generation quality. The method is mainly designed based on two insights. First, differently-sized models trained on the same data distribution share similar encodings thus one can switch between these models during the denoising steps. Second, although the smaller models have lower generation quality than larger models, they are sufficient in earlier denoising steps which generate image global structures. Thus, the proposed T-Stich method reduces the inference time by utilizing a smaller and faster model in the earlier steps and switches to the more capable but more expensive larger model in later steps and controls the trade-off between quality and speed by adjusting the fraction of steps using the large model. Experiments on different pre-trained DPM models show the proposed method can reduce the latency while maintaining generation quality. In addition, it shows using a general model in the early steps and a stylized model in later steps can provide better prompt alignment than completely using the stylized model.

**Strengths:**

S1. The proposed method is intuitive and simple. It is very easy to incorporate this method for any diffusion model as long as there are model variants of different sizes and inference costs that are trained on the same data distribution.

S2. The writing is easy to follow and the presentation is mostly clear.

S3. There is a good amount of experiments on different diffusion models and example images showing the proposed method can reduce the inference cost while maintaining comparable image quality when compared to the single model for all timesteps vanilla approach.

**Weaknesses:**

W1. This paper uses a too simple and strict design of using a weaker model in earlier steps and switching to a stronger model in later steps. However, there is not enough justification or comparison to other baseline approaches. How does it compare to more flexible baselines, e.g. interleaving strong model steps and weaker model steps throughout the whole process, or gradually reducing the probability p of using the weaker model e.g. from p=1 at t=T to and p=0 at t=0?

W2. Lack of comparison to other related works on multi-expert DPM approaches like [1] and [2]. Under the same inference time budget, how does the proposed approach compare to [1]? Does using a larger pre-trained model in the earlier steps and using a smaller model in later steps have better or worse generation quality compared to [1] which adopts differently designed architectures tailored toward the low-frequency features for the earlier steps or the high-frequency information for later denoising steps?

W3. In Figure 15, even the simple baseline of directly reducing the sampling steps outperforms the proposed method at the 10-50 steps range for s=1.5 and 10-20 steps range at s=2.0. This means the T-stitch approach could achieve better generation quality and latency tradeoff if combined with reducing steps and this was not investigated. More importantly, this result shows the proposed t-stitch approach does not have a strong performance even compared to this simple baseline and more comparisons to other approaches in the literature like [1] and [2] are needed.

References:
[1] Y. Lee, J.-Y. Kim, H. Go, M. Jeong, S. Oh, and S. Choi, ‘Multi-Architecture Multi-Expert Diffusion Models’,
[2] Y. Balaji et al., ‘ediffi: Text-to-image diffusion models with an ensemble of expert denoisers’

**Questions:**

Q1. Have you considered and compared to more flexible baselines, e.g. interleaving strong model steps and weaker model steps throughout the whole process, or gradually reducing the probability p of using the weaker model e.g. from p=1 at t=T to and p=0 at t=0?

Q2. Under the same inference time budget, how does the proposed approach compare to [1]? Does using a larger pre-trained model in the earlier steps and using a smaller model in later steps have better or worse generation quality compared to [1] which adopts differently designed architectures tailored toward the low-frequency features for the earlier steps or the high-frequency information for later denoising steps?

---

> ### Author Response · Authors · 2023-11-17
> **Author Response to Reviewer UhCD**
>
> Thanks for your valuable feedback. We would like to address your concerns as below.
>
> **Q1. Design is too simple, compared to other baselines.**
>
> We agree T-Stitch is a simple technique. However, it is also a generalizable and effective approach as highly recognized by other reviewers, i.e., “the insight …is clever….a simple yet novel idea…broadly applicable…highly practical” (Reviewer 9mJB ), “built upon the dynamics of diffusion models…clearly distinguishing it from model-wise stitching..” (Reviewer tbyv).
>
> Furthermore, in our initial experiments, we have explored various allocation strategies, such as interleaving and large-to-small, but found that our default design can achieve the best speed-quality trade-off. In the following, we demonstrate the detailed comparison with the mentioned baselines:
>
> 1. **Interleaving**. During denoising sampling, we interleave the small and large models along the trajectory, ie., DiT-S -> DiT-XL -> … -> DiT-S -> DiT-XL. Eventually, DiT-S takes 50% steps and DiT-XL takes another 50% steps.
> 2. **Decreasing Prob.** Linearly decreasing the probability of using DiT-S from 1 to 0 during the denoising sampling steps.
> 3. **Large-to-Small.** Adopting the large model at the early 50% steps and the small model at the last 50% steps.
> 4. **Small-to-Large (Ours).** The default strategy of T-Stitch by adopting DiT-S at the early 50% steps and using DiT-XL at the last 50% steps.
>
> All experiments are based on DiT-S and DiT-XL, with DDIM 100 steps and a guidance scale of 1.5. FID is calculated based on 5K images. As shown in the table below, under the similar time cost, our design achieves a superior advantage in FID and Inception Score compared to the other baselines. Nevertheless, we agree they are meaningful baselines, and we have included these comparisons in Section A.13 of the revised Appendix.
>
> | **Method**            | **FID↓**   | **Inception Score↑** | **Time Cost (s)** |
> | --------------------- | --------- | ------------------- | ----------------- |
> | Interleave            | 19.02     | 120.04              | 10.1              |
> | Decreasing Prob       | 12.94     | 163.45              | 9.8               |
> | Large-to-Small        | 27.61     | 72.60               | 10.0              |
> | Small-to-Large (Ours) | **10.06** | **200.81**          | 9.9               |
>
> > The time cost is measured by generating 8 images on one RTX 3090 in seconds (s).
>
> **Q2. Compared with MEME [1] and eDiff-I [2].**
>
> It is important to note that we are significantly different from MEME and eDiff-I:
>
> 1. T-Stitch accelerates **off-the-shelf pretrained** large diffusion models in a **training-free** manner, which is a general technique and enables “broad applicability”, as recognized by Reviewer 9mJB.
> 2. Existing multi-experts DPMs **design new models** and **train them from scratch** using expensive computing resources to either reduce the time cost per step with multiple smaller DPMs (MEME) or achieve better performance with multiple large DPMs without considering efficiency (eDiff-I).
>
> Essentially, as already discussed in the Remark of Section 3, we target the research problem of computing budget allocation across different steps while benefiting from **training-free**, while being directly applicable to SDs, Stylized SDs, SDXL (Figure 25) and ControlNet (Figure 26). Therefore, we are not directly comparable. Furthermore, at this time, MEME is an Arxiv preprint paper that does not release code/weights, thus might be inappropriate for us to compare the speed under the same hardware.
>
> **Q3. Comparing T-Stitch with reducing sampling steps, combining T-Stitch with fewer steps was not investigated.**
>
> In fact, T-Stitch allocates different compute budgets at different steps, which is a **complementary** technique with reducing sampling steps, **not competing with it.** Figure 15 additionally provides an ablation study of the direct comparison for comprehensive reference.
>
> More importantly, we think we **have already provided the experiments of combining T-Stitch with fewer steps** in Figure 8, Figure 9 and Figure 17. For example, under 10/20/50 steps, T-Stitch can maintain low FID (Figure 8 left) and high image quality (Figure 17) by replacing the early 40% steps with a small DiT-S to accelerate DiT-XL sampling, which clearly demonstrated that T-Stitch can achieve better speed-quality trade-offs if combined with reducing steps.

---

> ### Author Response · Authors · 2023-11-20
> **Follow-up Discussion**
>
> Dear Reviewer UhCD,
>
> We sincerely thank you again for your great efforts in reviewing our paper. We have provided responses to address your major concerns about the comparison with other trajectory stitching baselines, more discussion with related multi-expert DPMs [1] and [2], as well as the compatibility with reducing the sampling steps. Please don’t hesitate to let us know if you have any further questions.
>
> Best,
>
> Authors of Submission 901

---

> > ### Comment · Reviewer_UhCD · 2023-11-22
> >
> > Thank you for the additional results on comparison with other trajectory stitching baselines and clarification on compatibility with reducing the sampling steps. After reading the other reviews and the authors' responses, I believe that a comprehensive comparison with existing methods is needed and this concern is also raised by the AC and other reviewers.

---

> ### Author Response · Authors · 2023-11-22
> **Follow-up Discussion**
>
> Dear Reviewer UhCD,
>
> Thank you for raising your additional concern during the discussion. We have provided follow-up response to AC which briefly summarizes the relation between T-Stitch and existing acceleration methods. Please feel free to ask any further questions and we sincerely appreciate the opportunity to discuss with you.
>
> Best regards,
>
> Authors of Submission 901

---

### Official Review · Reviewer_omxW · 2023-11-01

**Soundness:** 3 good
**Presentation:** 3 good
**Contribution:** 2 fair
**Rating:** 5
**Confidence:** 4

**Summary:**

This paper proposes a accelerating sampling method of diffusion model. Based on the phenomenon that different diffusion models learn similar encodings under the same training data distribution, this paper proposes to use a small model in the early sampling period to learn the global structures, while a larger model being adopted in the later sampling period to learn high-frequency details.

**Strengths:**

1.	The proposed method can conveniently adopt the existing pretrained diffusion models without finetuning, to accelerate the sampling speed.
2.	The proposed method which using a small general expert at the beginning sampling stage of stable diffusion results in better prompt alignment.
3.	While a two-stage sampling is used in this paper, the proposed method can also be expanded to multi-stage.

**Weaknesses:**

1. Two models mean more storage consume, or if they are sent into the GPU in order, they will be in and out for every batch, which is not convenient.
2. The authors are recommended to compare the speed of their proposed method with the other accelerating methods mentioned in the second paragraph of Introduction.

**Questions:**

Please refer to my comments on weaknesses.

---

> ### Author Response · Authors · 2023-11-17
> **Author Response to Reviewer omxW**
>
> Thanks for your efforts for reviewing our paper. We would like to address your concerns as below.
>
> **Q1. Two models mean more storage consumption, and it is not convenient on GPUs.**
>
> It is worth noting that compared to the large model, a small model only **slightly increases** the memory (x1.04) and local disk storage (x1.1), as shown in our general response Part-2.
>
> Besides, **T-Stitch is actually very efficient on GPUs** **since both models are pre-loaded into GPU memory before inference.** During the denoising sampling, model switching is equivalent to choosing a different computational graph immediately, thus resulting in a minor overhead.
>
> For example, as shown in our response to Reviewer UhCD Q1, based on DiT-S/XL, even interleaving DiT-S and DiT-XL during the sampling trajectory takes the similar time cost (10.1s) as our default strategy (9.9s) of adopting DiT-S for the early 50% steps then switch into DiT-XL for the last 50% steps, which is still clearly faster than only using DiT-XL for sampling (16.5s).
>
> **Q2. Compared to other acceleration techniques**
>
> In fact, we are not competing with single models/samplers/reducing steps. Instead, T-Stitch efficiently adopts pretrained small DPMs as cheap drop-in-replacements for large DPMs during sampling. Thus, we are **complementary to other sampling acceleration techniques,** which has been extensively discussed in the related works, as also recognized by both Reviewer 9mJB (*“...complements existing techniques.”*) and Reviewer tbyv (*“...complementary to advanced diffusion samplers based on better ODE discretization.”*). More importantly, we have already demonstrated this compatibility with comprehensive experiments on different number of sampling steps, samplers and architectures in Figure 8, Figure 9 and Table 1, respectively.

---

> ### Author Response · Authors · 2023-11-20
> **Follow-up Discussion**
>
> Dear Reviewer omxW,
>
> We sincerely thank you again for your considerable efforts in reviewing our paper. We have provided responses to address your major concerns about the storage overhead, the efficiency on GPUs, as well as the compatibility with other acceleration techniques.  Please feel free to reach out if you have any further questions.
>
> Best,
>
> Authors of Submission 901

---

> > ### Comment · Reviewer_omxW · 2023-11-21
> >
> > Thanks for your rebuttal. As for an accelerating technique, I think giving an comprehensive comparsion with existing methos is necessary for evaluating a new method.

---

> > > ### Author Response · Authors · 2023-11-22
> > > **Follow-up Response**
> > >
> > > Dear Reviewer omxW,
> > >
> > > Thanks for your valuable time in engaging the discussion. We have provided detailed explanations and new results in the follow-up response to AC. Please don’t hesitate to let us know if you have any further questions.
> > >
> > > Best regards,
> > >
> > > Authors of Submission 901

---

### Official Review · Reviewer_tbyv · 2023-11-05

**Soundness:** 3 good
**Presentation:** 3 good
**Contribution:** 3 good
**Rating:** 8
**Confidence:** 4

**Summary:**

This paper introduced trajectory stitching (T-Stitch), a simple approach to accelerate the sampling process of diffusion models by dynamically allocating computations to the sampling trajectory. The motivation for this work was the observation from prior works that different DPMs trained in the same data distribution learn similar score estimation regardless of model sizes and architectures. Further investigations show the frequency bias of diffusion models at varying noise levels. Altogether, this motivates this work to stitch the early sampling trajectory from smaller models with ones of larger models, where smaller models and larger models correspond to global shape and local textures, respectively.

The proposed technique accelerates the sampling speed by 40% w/o quality degradation or retraining. It is also complementary to advanced diffusion samplers based on better ODE discretization. Surprisingly, T-Stitch improves the prompt alignment of stylized latent diffusion models (LDMs).

**Strengths:**

This work has clear merits in its motivation and easy-to-understand simple technique. I enjoy the clarity of writing. It is also reminiscent of speculative decoding for language models. Importantly, this stitching is built upon the dynamics of diffusion models, clearly distinguishing it from model-wise stitching and being off-the-shelf for pretrained models. The experiments show the Pareto frontier produced by T-stitch and its advantage over the baseline setup.

**Weaknesses:**

However, the drawbacks of this work are also apparent. Despite improving the prompt alignment of stylized Stable Diffusion (SD) models, there needs to be a clear investigation into why this could happen. It demonstrated clever empirical usage of prior observations but still failed to dig into the phenomena to offer better depth and insights.

The technique drawback, although preventing it from reaching more elevated quality, is not a barrier to accepting this work. I'd agree that the current scope has met the bar of ICLR. Good work!

**Questions:**

Please see above.

---

> ### Author Response · Authors · 2023-11-17
> **Author Response to Reviewer tbyv**
>
> Thanks for your very positive comments! In addition to the initial discussion in Section 4.3, we would like to provide further analysis as below,
>
> 1. **Dreambooth finetuning is easy to overfit.** Stylized SD models are usually finetuned on a very limited number of images, which is easy to result in overfitting and catastrophic forgetting [A, B]. For example, the Ghibili/Inkpunk Diffusion models in Figure 2 are finetuned with DreamBooth. However, according to the official documentation on Huggingface, “DreamBooth finetuning is very sensitive to hyperparameters and easy to overfit ” [C], which may inevitably result in a loss in terms of the pretrained general knowledge [D].
> 2. **General SD may complement the knowledge.** The small SD (BK-SDM Tiny) is a pruned and distilled version of the original SD v1.4, which could preserve the majority of the generality. Therefore, by adopting the small SD at the early steps, it may provide some general priors at the beginning [E], thus complementing the missing concepts in the prompts for the overfitted stylized SD models.
>
> We agree that more in-depth analysis can be helpful to understand the prompt alignment for stylized SD models, but as our main aim is to achieve acceleration, we will leave such exploration for future work.
>
> [A] Zhang, Lvmin, Anyi Rao, and Maneesh Agrawala. "Adding conditional control to text-to-image diffusion models." *ICCV*. 2023.
>
> [B] Ruiz, Nataniel, et al. "Dreambooth: Fine tuning text-to-image diffusion models for subject-driven generation." *CVPR*. 2023.
>
> [C] https://huggingface.co/docs/diffusers/training/dreambooth#finetuning
>
> [D] Lin, Yong, et al. "Speciality vs generality: An empirical study on catastrophic forgetting in fine-tuning foundation models." *arXiv* (2023).
>
> [E] Graikos, Alexandros, et al. "Diffusion models as plug-and-play priors." *NeurIPS* (2022).

---

> > ### Comment · Reviewer_tbyv · 2023-11-20
> >
> > Thanks for the response. I am glad to see these as helpful additions to the manuscript. :)

---

> > > ### Author Response · Authors · 2023-11-20
> > > **Thank You!**
> > >
> > > Dear Reviewer tbyv,
> > >
> > > Thanks for your feedback! We are pleased to address your concerns and are grateful for the significant role your reviews play in improving our work.
> > >
> > > Best,
> > >
> > > Authors of Submission 901

---

### Official Review · Reviewer_9mJB · 2023-11-09

**Soundness:** 3 good
**Presentation:** 3 good
**Contribution:** 3 good
**Rating:** 6
**Confidence:** 3

**Summary:**

In summary, T-Stitch is a simple yet effective way to accelerate sampling in large diffusion models by strategically combining them with smaller models, with little or no drop in sample quality. The results demonstrate it is broadly applicable across model architectures.

- It proposes a method called "Trajectory Stitching" (T-Stitch) to accelerate sampling in pretrained diffusion models without loss of quality. The key idea is to use a smaller, faster diffusion model for the initial sampling steps and switch to a larger, higher quality model later in the process.
- It is based on the observation that different diffusion models trained on the same data distribution learn similar latent representations, especially in early sampling steps. So the small model can handle the initial coarse sampling while the large model refines details later.
Experiments show T-Stitch can accelerate sampling in various diffusion architectures like DALL-E, Stable Diffusion, etc without quality loss. For example, with DiT models it allows replacing 40% of steps with a 10x faster model without performance drop on ImageNet.
T-Stitch also improves prompt alignment in finetuned diffusion models like stable diffusion. This is because finetuning can hurt prompt alignment which the small general model can complement.
- The method is complementary to other sampling acceleration techniques like model compression, distillation etc. Those can be applied to the part handled by the large model.
- T-Stitch achieves better speed vs quality tradeoffs compared to model stitching techniques like SN-Net which permanently stitch model components. T-Stitch stitches sampling trajectories.

**Strengths:**

Overall, T-Stitch demonstrates a simple, generalizable, and effective approach for diffusion sampling acceleration that complements existing techniques. The strong experimental results and ablation analysis make a compelling case for the method.
- Novel Idea: Trajectory stitching is a simple yet novel idea of accelerating diffusion sampling by combining models of different sizes. Prior work mostly focused on using a single model. The insight of leveraging similarity in early sampling latents is clever.
- Broad Applicability: The method is shown to be broadly applicable across various diffusion model architectures like DALL-E, Stable Diffusion, U-Nets etc. It also improves finetuned models like stylized Stable Diffusion. This demonstrates the generality of the approach.
- Pareto Optimality: T-Stitch provides better speed vs quality tradeoffs compared to techniques like model stitching and even some training based methods. The Pareto frontier is improved.
- Realistic Setting: The method is evaluated in realistic settings using widely adopted models like Stable Diffusion. Showing acceleration and prompt alignment improvement makes it highly practical.

**Weaknesses:**

- Memory Overhead: Adopting additional smaller models during sampling increases memory usage, which could be a concern for very large models.
- Finicky Tuning: Getting the right model stitching fractions to optimize the speed-quality tradeoffs may require finicky tuning based on the models and datasets. More principled guidelines could help.
- Theoretical Analysis: While FID evaluates sample quality well, measuring sample diversity could be helpful to ensure stitching does not negatively impact it. The paper lacks theoretical analysis and justification on why stitching trajectories preserves sample quality, beyond empirical evidence.

**Questions:**

1. For finetuning experiments, can you elaborate on the exact finetuning procedure? Was it only on stitched intervals?  How do fully finetuned models compare?
2. The prompts used for stable diffusion examples are quite simple. Have you tried more complex prompts and datasets? How robust is the method?

---

> ### Author Response · Authors · 2023-11-17
> **Author Response to Reviewer 9mJB - Part-1**
>
> Thanks for your constructive and positive comments! We would like to address your additional concerns as follows.
>
> **Q1. Memory Overhead of T-Stitch.**
>
> We agree T-Stitch slightly increases memory usage by adopting a smaller model, as already stated in the limitations. However, in practice, this overhead can be very minor since the main bottleneck still lies in the large models. Please refer to our general response Part-2 for more details.
>
> **Q2. Finicky tuning of getting the right T-Stitch fractions to optimize the speed-quality trade-offs.**
>
> In practice, T-Stitch naturally interpolates a smooth speed-quality curve between a small and large DPM under different fractions of the small DPM. As this curve generally exists under different samplers (Figure 9), numbers of steps (Figure 8), and architectures (Tables 1,2,5), the speed under a given fraction can be roughly estimated based on the almost linearly interpolated time cost curve between the two DPMs (Figure 8 Right), while the more accurate quality measurement can be obtained by querying a pre-computed look-up table, as discussed in Section A.1 of the initial submission. We also agree with the reviewer that more principled guidelines would be helpful and will leave it for future work.
>
> **Q3. Diversity measurement of T-Stitch.**
>
> Following common practice [A], we adopt Precision to measure fidelity, and Recall to measure diversity or distribution coverage. In the table below, we report the results based on DiT-S/XL, 100 DDIM steps, and a guidance scale of 1.5. As it shows, T-Stitch maintains high Precision and Recall at the early 40-50% steps, which is consistent with FID evaluations. We agree diversity measurement is also important thus we have included this result in Section A.15 of the revised Appendix.
>
> | Fraction of DiT-S | 0%   | 10%  | 20%  | 30%  | 40%  | 50%  | 60%  | 70%  | 80%  | 90%  | 100% |
> | ----------------- | ---- | ---- | ---- | ---- | ---- | ---- | ---- | ---- | ---- | ---- | ---- |
> | Precision ↑       | 0.81 | 0.81 | 0.81 | 0.81 | 0.80 | 0.76 | 0.72 | 0.67 | 0.62 | 0.59 | 0.58 |
> | Recall ↑          | 0.74 | 0.74 | 0.74 | 0.74 | 0.75 | 0.75 | 0.74 | 0.73 | 0.69 | 0.65 | 0.63 |
>
> [A] Dhariwal, Prafulla, and Alexander Nichol. "Diffusion models beat gans on image synthesis." *NeurIPS* (2021): 8780-8794.
>
> **Q4. Theoretical analysis and justification on why stitching trajectories preserves sample quality.**
>
> Thanks for pointing this out. We agree that more in-depth theoretical proof is missing at this stage. In addition to our initial justifications in the first paragraph of Section 3.2, we would like to provide further analysis as follows,
>
> Theoretically, under the same dataset, score-based models trained with the same denoising score matching loss aim to learn a consistent score function [B], which guides the probability flow at the reverse time to produce the overall ODE trajectory. If they learn the same ground-truth score, they will have the same reverse trajectory. This is evidenced by our Figure 3 where different DiTs learn similar intermediate latent embeddings, thus indicating a similar trajectory. Essentially, T-Stitch forwards the latent codes from the last timestep of a small model into the current timestep for the large model, which provides roughly similar scores, thus the stitched trajectory may interpolate a similar ODE trajectory and produce similar images, as shown in Figure 17.
>
> Due to the current scope, we will leave more comprehensive analysis for future work.
>
> [B] Song, Yang, et al. "Score-based generative modeling through stochastic differential equations." *ICLR* (2021).

---

> ### Author Response · Authors · 2023-11-17
> **Author Response to Reviewer 9mJB - Part-2**
>
> **Q5: Finetuning experiments**
>
> In Section A.12, we separately finetune the pretrained DiT-B and DiT-XL **at their stitched intervals**. Concretely,
>
> 1. During finetuning DiT-B, at each training iteration, we limit the timesteps to be only sampled from the intermediate 30% steps.
> 2. Similarly, during finetuning DiT-XL, we only sample the timesteps from the last 20% steps.
> 3. After both models are finetuned with 250K iterations, we directly apply T-Stitch based on the denoising intervals of DiT-S/B/XL with 50% : 30% : 20%.
>
> To compare with finetuning at all intervals, we finetuned the pretrained DiT-B/XL for **all timesteps** with the same additional 250K iterations. The table below evaluates the same T-Stitch allocation (50% : 30% : 20%) and demonstrates that this strategy does not yield superior performance, suggesting that finetuning at stitched intervals is more effective.
>
> | **Method**                     | **FID ↓** | **Inception Score ↑** |
> | ------------------------------ | --------- | --------------------- |
> | Pretrained                     | 16.49     | 123.11                |
> | Finetuned at all timesteps     | 16.04     | 125.81                |
> | Finetuned at stitched interval | 13.35     | 155.35                |
>
> **Q6: More complex prompts and datasets for stable diffusion examples.**
>
> Thanks for the suggestion! We have included more examples by using more complex prompts in the Appendix Figure 24 and Figure 25, where T-Stitch performs favorably with long and complex prompts, and can be **easily applied into SDXL and ControlNet (Figure 26)**. Moreover, we write a for-loop script for generating image samples with **8 consecutive runs** under the same prompts but different latent noises. As shown in Figure 27, the output images by adopting different fractions of the small SD show great image quality for all runs. Thus it indicates T-Stitch is robust in practice.

---

> ### Author Response · Authors · 2023-11-20
> **Follow-up Discussion**
>
> Dear Reviewer 9mJB,
>
> We sincerely thank you again for your great efforts in reviewing our paper. We have provided responses to address your major concerns about the memory overhead, tuning guidelines and additional analysis on T-Stitch. Beside that, we have also enriched the section for the finetuning experiment and shown more stable diffusion examples under complex prompts. Please don’t hesitate to let us know if you have any further questions.
>
> Best,
>
> Authors of Submission 901

---

### Author Response · Authors · 2023-11-17
**General Response**

We sincerely thank all the reviewers for their insightful feedback and would like to provide a summary of the rebuttal and some additional clarifications.

## 1. Summary of reviews

In general, T-Stitch is highly recognized by most reviewers,

- “...novel idea, the insight … is clever, broadly applicable, better speed vs quality tradeoffs, and highly practical”, “...complementary to other sampling acceleration techniques” (Reviewer 9mJB),
- “has clear merits in its motivation…being off-the-shelf for pretrained models…good work” (Reviewer tbyv),
- “conveniently adopt the existing pretrained diffusion models without finetuning” (Reviewer omxW)
- “...very easy to incorporate this method for any diffusion model…good amount of experiments” (Reviewer UhCD)

Following the additional concerns raised by reviewers, we have provided detailed responses in the rebuttal.

## 2. Additional memory and storage overhead of T-Stitch

We would like to point out that the main bottlenecks on GPU memory and local disk storage are still from the large models. The additional overhead of T-Stitch can be considerably minor. As shown in the table below, adopting DiT-S only introduces additional 5% parameters, 4% GPU memory cost, 10% local storage cost, while significantly accelerating DiT-XL sampling speed by 1.76x.

|                | **Parameter (M)** | **Local Storage (MB)** | **Memory Cost (MB)** | **Time Cost (s)** |
| -------------- | ----------------- | ---------------------- | -------------------- | ----------------- |
| DiT-S          | 33                | 263                    | 3,088                | 1.70              |
| DiT-XL         | 675               | 2,576                  | 3,166                | 16.5              |
| T-Stitch (50%) | 708 (x1.05)       | 2,839 (x1.10)          | 3,296 (x1.04)        | 9.4 (x1.76)       |

> Memory and time cost are measured by a batch size of 8 on one RTX 3090. We report the peak memory consumption via `torch.cuda.max_memory_allocated()` during the denoising sampling process. We adopt a fraction of 50% in T-Stitch, i.e., allocating DiT-S for the early 50% steps and DiT-XL for the last 50% steps. For other fractions, we observe similar memory costs.

## 3. Summary of paper revision

According to the feedback from the reviewers, we have revised/included the following sections and results,

1. **(Revision)** We have enriched the details for the finetuning experiments in Section A.11.
2. **(New)** We have included the comparison with more baselines of different trajectory stitching strategies in Section A.13.
3. **(New)** We report the additional overhead of memory and storage cost by T-Stitch in Section A.14.
4. **(New)** We report the Precision and Recall results based on DiT-S/XL in Section A.15.
5. **(New)** In Figure 24, we show **more examples with complex prompts** for Stable Diffusion V1.4.
6. **(New)** In Figure 25 and Figure 26, we show T-Stitch can **easily accelerate SDXL 1.0** with a smaller and distilled version of SSD-1B, while being **directly applicable to ControlNet** for practical art generation under complex prompts.
7. **(New)** In Figure 27, we show the image examples under T-Stitch by 8 consecutive runs.

We thank the reviewers and ACs again for their efforts in reviewing our paper, and sincerely welcome further discussions.

Best regards,

Authors of Submission 901

---

### Comment · Area_Chair_EGP3 · 2023-11-21
**Discussion is needed**

Reviewers: Opinions on the paper vary quite a bit. In addition to considering the authors' rebuttal, could you also review the feedback provided by the other reviewers?

Authors: Model stitching, as described in Appendix A.8, is positioned as a notable baseline, yet its standing as a widely accepted and benchmarked technique for accelerated diffusion sampling remains uncertain. The introduction of model stitching prompts the question of why T-stitch has not been compared to other established learning/finetuning-based acceleration methods, such as those rooted in distillation or hybrid modeling (e.g., VAE or GAN stitched with Diffusion).

Thanks,

AC

---

> ### Author Response · Authors · 2023-11-22
> **Follow-up Response**
>
> Dear AC,
>
> Thanks for your valuable time to join the discussion and read our rebuttal. We appreciate the opportunity to address your concerns as follows:
>
> **1. Why introduce model stitching as a notable baseline?**
>
> Our insight is that *“using the same model throughout is a suboptimal strategy for efficiency”,* thus we aim to explore more flexible trade-offs between exclusively adopting a large DPM or a small DPM for sampling. To this end, model stitching [A, B] is an effective framework that obtains numerous architectures with different speed and quality trade-offs given a pretrained model family, which provides an initial appropriate baseline that aligns with our objective. However, most existing acceleration techniques focus on individual model optimization by distillation, pruning or redesign which still suffer from fixed model complexity, making it difficult for us to study the problem of compute budget allocation during the sampling process. Model stitching, therefore, is instrumental in showcasing the superiority of T-Stitch in this case.
>
> **2. Why has T-stitch not been directly compared to other established learning/finetuning-based acceleration methods?**
>
> T-Stitch is a **training-free** technique for obtaining flexible trade-offs between a pretrained small and a pretrained large DPM, which might be inappropriate to directly compare with other **training-based** methods. Essentially, our acceleration is driven by the small DPM, where the faster and better it is, the better speed-quality trade-off we can achieve, thus it can be **orthogonal** to the choice of the large DPMs (eg, DiT-S/XL in Figure 5 and DiT-S/U-ViT-H in Table 5), the number of sampling steps (Figure 8), and samplers (Figure 9).
>
> On the other hand, T-Stitch can further **speed up an already accelerated DPM via established training-based methods**, such as distillation. For example, as shown in Figure 28 and Figure 29 (**new results**), given a distilled SDXL from LCM [C] (a concurrent work), T-Stitch can achieve further speedup under 2~4 steps with high image quality by adopting a relatively smaller SD. For example, under 4 steps, applying T-Stitch (75%) reduces the sampling time of LCM distilled SDXL by 15% without sacrificing visual quality. These results demonstrate that T-stitch is generally applicable across various scenarios. Hybrid modeling such as Diffusion-GAN [D] aims to improve GAN and also needs training, which is beyond our scope of diffusion model acceleration at this stage.
>
> We thank the AC again for the efforts in reviewing our paper, and sincerely welcome further discussions.
>
> Best regards,
>
> Authors of Submission 901
>
> [A] X, Yang, et al. "Deep model reassembly." NeurIPS 2022.
>
> [B] Z, Pan, et al. "Stitchable Neural Networks." CVPR 2023.
>
> [C] S, Luo, et al. "Latent consistency models: Synthesizing high-resolution images with few-step inference." arXiv preprint arXiv:2310.04378, 2023.
>
> [D] Z, Wang, et al. "Diffusion-gan: Training gans with diffusion." ICLR 2023.

---

> > ### Comment · Area_Chair_EGP3 · 2023-11-22
> >
> > Thank you for thoroughly addressing my comments. I understood that T-stitch is training-free, providing practical utility. I was hoping for a more extensive comparison with fine-tuning/learning-based baselines, extending beyond model stitching, to better evaluate the value of T-stitch. The demonstration of compatibility with distilled large models is a positive development. I believe there are clear understandings of the strengths and weaknesses of the paper, and I will discuss these with the reviewers to reach a final decision.

---

> ### Author Response · Authors · 2023-11-22
> **Follow-up Response - Part 2**
>
> Dear AC,
>
> Thanks for your further comments. Beside that aforementioned clarifications, we would like to provide a brief summary on the relation between T-Stitch and existing DPM acceleration techniques and hopefully provide a clear big picture.
>
> Previous fast sampling methods for diffusion models, including training-free and training-based ones, are mainly focusing on reducing sampling steps while our approach mainly targets for reducing the overall sampling time under the same sampling steps. This major difference makes T-Stitch completely complementary to previous methods. More discussions are available in the introduction and related work sections. Because our method does not need training at all, it serves as a simple drop-in accelerator that is compatible with existing training-free and training-based methods. And we have shown the compatibility with acceleration techniques as follows:
>
> - Directly reducing sampling steps (Figure 8, Figure 17)
> - Advanced samplers (Figure 9)
> - Model compression (Figure 16)
> - Distillation for sampling with fewer steps (Figure 28, Figure 29).
>
> We have also demonstrated that T-Stitch is broadly applicable to various scenarios,
>
> - Transformer (Figure 5) and U-Net (Table 1).
> - Different pretrained model families, i.e., DiT + U-ViT (Table 5)
> - Adopting more models for more flexible trade-offs (Figure 6)
> - Pretrained SDs (Table 2 and Figure 21)
> - Stylized SDs and potential better prompt alignment (Figure 22, Figure 23)
> - Complex Prompts and SDXL (Figure 24, Figure 25)
> - ControlNet (Figure 26)
>
> Overall, with comprehensive experiments (*"strong experimental results and ablation analysis"* from Reviewer 9mJB and "good amount of experiments" from Reviewer UhCD), we have shown T-Stitch offers all its benefits without the need for training, serving as a general tool that can be readily applied to existing DPM acceleration techniques for achieving further speedup, while maintaining comparable performance.
>
> We sincerely thank the AC and reviewers again for the valuable time in engaging the discussion.
>
> Best regards,
>
> Authors of Submission 901

---

### Meta-Review · Area_Chair_EGP3 · 2023-12-08

**Metareview:**

The paper builds upon prior research findings, which observed that different diffusion probabilistic models (DPMs) trained on the same data distribution tend to learn similar score estimation, regardless of their model sizes and architectures. The paper's main proposition is to initiate sampling with a smaller model and then transition to a larger one to expedite the sampling process. The paper presents a training-free approach and supports its ideas with extensive experimental evidence. While the paper leverages previous observations about DPM behaviors, it doesn't introduce clear innovations in theory or methods. To enhance its value, it would benefit from a comparative analysis with well-established acceleration techniques, including training-based methods.

**Justification For Why Not Higher Score:**

It doesn't introduce clear innovations in theory or methods. It would benefit from a comparative analysis with well-established acceleration techniques, including training-based methods.

**Justification For Why Not Lower Score:**

N/A

---

### Decision · Program_Chairs · 2024-01-16

Reject